# Exploring Symmetry of Binary Classification Performance Metrics

**Amalia Luque [1],*** , **Alejandro Carrasco [2]** , **Alejandro Martín [1]** and **Juan Ramón Lama [1]**

[1]  Ingeniería del Diseño; Escuela Politécnica Superior. Universidad de Sevilla, 41011 Sevilla, Spain; ammartin@us.es (A.M.); jrlama@us.es (J.R.L.)
[2]  Tecnología Electrónica; Escuela Ingeniería Informática. Universidad de Sevilla, 41012 Sevilla, Spain; acarrasco@us.es
*  Correspondence: amalialuque@us.es; Tel.: +34-955-420-187

**Abstract:** Selecting the proper performance metric constitutes a key issue for most classification problems in the field of machine learning. Although the specialized literature has addressed several topics regarding these metrics, their symmetries have yet to be systematically studied. This research focuses on ten metrics based on a binary confusion matrix and their symmetric behaviour is formally defined under all types of transformations. Through simulated experiments, which cover the full range of datasets and classification results, the symmetric behaviour of these metrics is explored by exposing them to hundreds of simple or combined symmetric transformations. Cross-symmetries among the metrics and statistical symmetries are also explored. The results obtained show that, in all cases, three and only three types of symmetries arise: labelling inversion (between positive and negative classes); scoring inversion (concerning good and bad classifiers); and the combination of these two inversions. Additionally, certain metrics have been shown to be independent of the imbalance in the dataset and two cross-symmetries have been identified. The results regarding their symmetries reveal a deeper insight into the behaviour of various performance metrics and offer an indicator to properly interpret their values and a guide for their selection for certain specific applications.

**Keywords:** performance metrics; classification; computational symmetry; machine learning

## 1. Introduction

Symmetry has played and continues playing, a highly significant role in the way of how humans perceive the world [1]. In the scientific fields, symmetry plays a key role as it can be discovered in nature [2,3], society [4] and mathematics [5]. Moreover, symmetry also provides an intuitive way to attain faster and deeper insights into scientific problems.

In recent years, an increasing interest has arisen in detecting and taking advantage of symmetry in various aspects of theoretical and applied computing [6]. Several studies involving symmetry have been published in network technology [7], human interfaces [8], image processing [9], data hiding [10] and many other applications [11].

On the other hand, pattern recognition and machine learning procedures are becoming key aspects of modern science [12] and the hottest topics in the scientific literature on computing [13]. Furthermore, in this field, symmetry is playing an interesting role either as a subject of study, in the form of machine learning algorithms to discover symmetries [14] or as a means to improve the results obtained by automatic recognition systems [15]. Let us emphasize this point: not only can knowing the symmetry of a certain computer algorithm be intrinsically rewarding since it sheds light on the behaviour of the algorithm but it can also be very useful for its interpretation, its optimization or as a

criterion for the selection among various competing algorithms. As an example, in recent research, we have employed a symmetric criterion to select the best feature-extraction procedures (Discrete Cosine Transform versus Discrete Fourier Transform) [16] in an application of the classification of sounds [17,18] effectively deployed in a Wireless Sensor Network as shown in Figure 1. Another examples of industrial applications using classification of sounds can be found in Refs. [19,20].

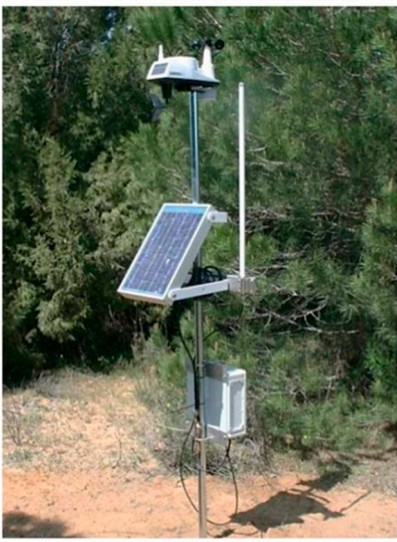

**Figure 1.** Node of the Wireless Sensor Network where the symmetry of classification performance metrics has been primarily applied.

In the broad field of machine learning, the study of how to measure the performance of various classifiers has attracted continued attention [21–23]. Classification performance metrics play a key role in the assessment of the overall classification process in the test phase, in the selection from among various competing classifiers in the validation phase and are even sometimes used as the loss function to be optimized in the process of model construction during the classification training phase.

However, to the best of our knowledge, no systematic study into the symmetry of these metrics has yet been undertaken. By discovering their symmetries, we would reach a better understanding of their meaning, we could obtain useful insights into when their use would be more appropriate and we would also gain additional and meaningful indicators for the selection of the best performance metric.

Although several dozen performance metrics can be found in the literature, we will focus on those which are probably the most commonly used: the metrics based on the confusion matrix [24]. Accuracy, precision and recall (sensitivity) are undoubtedly some of the most popular metrics. On the other hand, our research will be focused on the cases where there are only two classes (binary classifiers). Although this is certainly a limitation, it does provide a solid ground base for further research. Moreover, multiclass performance metrics are usually obtained by decomposing the multiclass problem into several binary classification sub-problems [25].

## 2. Materials and Methods

### 2.1. Definitions

Let us first consider an original (baseline) experiment $E^B$, defined by the duple $E^B = (C^B, D^B)$ composed of a set of $n^B$ classifiers, $C^B = \{c_i^B\}$ and a set of their corresponding $n^B$ datasets, $D^B = \{D_i^B\}$, $i = 1, \cdots, n^B$. The elements in every dataset belong to either of two classes, $G_1$ and $G_2$, which are called Positive ($P$) and Negative ($N$) classes, respectively. The $i$-th classifier $c_i^B$ operates on the corresponding $D_i^B$ dataset, thereby obtaining a resulting classification which can be defined by its binary confusion matrix $cm_i^B$ and hence $D_i^B \xrightarrow{c_i^B} cm_i^B$. The set of confusion matrices are denominated

$CM^B = \left\{ cm_i^B \right\}$. The baseline experiment can therefore be defined as the set of classifiers operating on the set of datasets to obtain a set of confusion matrices, $E^B : D^B \xrightarrow{C^B} CM^B$ .

This paper will explore the behaviour of binary classification performance metrics when the original experiment is subject to $n^E$ different types of transformations. Let us define the $k$-th transformed experiment $E^k = \left( C^k, D^k \right)$ composed of a set of $n^k$ classifiers, $C^k = \left\{ c_i^k \right\}$ and a set of their corresponding $n^k$ datasets, $D^k = \left\{ D_i^k \right\}$, whose result is a set of confusion matrices $CM^k = \left\{ cm_i^k \right\}$. Hence, $E^k : D^k \xrightarrow{C^k} CM^k$, where $k = \left\{ B, 1, 2, \cdots, n^E \right\}$, indicates the type of transformation. In the $k$-th experiment, when the $i$-th classifier $c_i^k$ operates on its corresponding $D_i^k$ dataset, the result is summarized in the binary confusion matrix defined as

$$cm_i^k = \begin{bmatrix} a_i^k & f_i^k \\ g_i^k & b_i^k \end{bmatrix}, \tag{1}$$

where

- $a_i^k$ is the number of positive elements in $D_i^k$ correctly classified as positive;
- $b_i^k$ is the number of negative elements in $D_i^k$ correctly classified as negative;
- $f_i^k$ is the number of positive elements in $D_i^k$ incorrectly classified as negative; and
- $g_i^k$ is the number of negative elements in $D_i^k$ incorrectly classified as positive.

Let us call $P_i^k$, $N_i^k$ and $M_i^k$ the positive, negative and total number of elements in $D_i^k$. Therefore $M_i^k = P_i^k + N_i^k$, $a_i^k + f_i^k = P_i^k$ and $g_i^k + b_i^k = N_i^k$. The confusion matrix can then be described as

$$cm_i^k = \begin{bmatrix} a_i^k & P_i^k - a_i^k \\ N_i^k - b_i^k & b_i^k \end{bmatrix}. \tag{2}$$

Let us now define $\alpha_i^k$ as the ratio of positive elements in $D_i^k$ correctly classified as positive; and $\beta_i^k$ as the ratio of negative elements in $D_i^k$ correctly classified as negative. That is,

$$\alpha_i^k \equiv \frac{a_i^k}{P_i^k}, \quad \beta_i^k \equiv \frac{b_i^k}{N_i^k}. \tag{3}$$

The confusion matrix can therefore be rewritten as

$$cm_i^k = \begin{bmatrix} \alpha_i^k P_i^k & P_i^k - \alpha_i^k P_i^k \\ N_i^k - \beta_i^k N_i^k & \beta_i^k N_i^k \end{bmatrix} = \begin{bmatrix} \alpha_i^k P_i^k & \left(1 - \alpha_i^k\right) P_i^k \\ \left(1 - \beta_i^k\right) N_i^k & \beta_i^k N_i^k \end{bmatrix}. \tag{4}$$

On the other hand, a dataset $D_i^k$ is called imbalanced if it has a different number of positive and negative elements, that is, $P_i^k \neq N_i^k$. Classification on the presence of imbalanced datasets is a challenging task requiring specific considerations [26]. To quantify the imbalance, several indicators have been proposed, such as the dominance [27,28], the proportion between positive and negative instances (formalized as $1 : X$) [29] and the imbalance ratio ($IR$) defined as $P_i^k / N_i^k$ [30], which is also called skew [31]. This value lies within the $[0, \infty]$ range and has a value $IR = 1$ in the balanced case. We prefer to use an indicator showing a value 0 in the balanced case, a value $+1$ when all the elements in the dataset are positive and $-1$ if all the elements are negative. We define the imbalance coefficient $\delta_i^k$, which is an indicator that has these characteristics, as

$$\delta_i^k \equiv 2 \frac{P_i^k}{M_i^k} - 1. \tag{5}$$

The imbalance coefficient is graphically shown in Figure 2 (solid blue cline) as a function of the proportion of positive elements in the dataset. For the sake of comparison, that figure also shows the *IR* imbalance ratio (dashed green line).

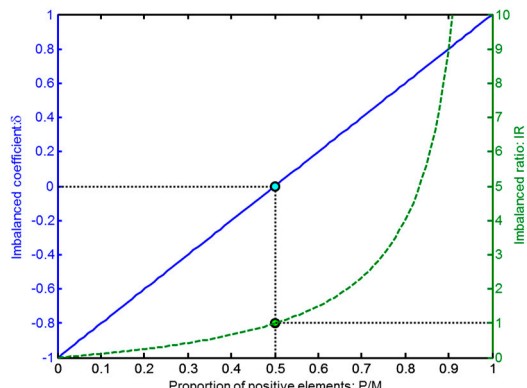

**Figure 2.** Imbalance coefficient (solid blue line) and imbalance ratio (dashed green line) vs. the proportion of positive elements in the dataset.

Based on the imbalance coefficient, the number of positive and negative elements in the dataset can be rewritten as

$$P_i^k = \frac{1 + \delta_i^k}{2} M_i^k. \tag{6}$$

$$N_i^k = M_i^k - P_i^k = M_i^k \left( 1 - \frac{1 + \delta_i^k}{2} \right) = \frac{1 - \delta_i^k}{2} M_i^k. \tag{7}$$

By substituting these expressions into Equation (4), the confusion matrix becomes

$$cm_i^k = \begin{bmatrix} \alpha_i^k \frac{1+\delta_i^k}{2} M_i^k & \left(1 - \alpha_i^k\right) \frac{1+\delta_i^k}{2} M_i^k \\ \left(1 - \beta_i^k\right) \frac{1-\delta_i^k}{2} M_i^k & \beta_i^k \frac{1-\delta_i^k}{2} M_i^k \end{bmatrix} = \lambda_i^k M_i^k, \tag{8}$$

where $\lambda_i^k$ is the unitary confusion matrix defined as

$$\lambda_i^k \equiv \begin{bmatrix} \alpha_i^k \frac{1+\delta_i^k}{2} & \left(1 - \alpha_i^k\right) \frac{1+\delta_i^k}{2} \\ \left(1 - \beta_i^k\right) \frac{1-\delta_i^k}{2} & \beta_i^k \frac{1-\delta_i^k}{2} \end{bmatrix}. \tag{9}$$

It can be seen that $\lambda_i^k$ is a function of 3 variables: the ratio of positive $\left( \alpha_i^k \right)$ and negative $\left( \beta_i^k \right)$ correctly classified elements and the imbalance coefficient $\left( \delta_i^k \right)$, that is, $\lambda_i^k = \lambda_i^k \left( \alpha_i^k, \beta_i^k, \delta_i^k \right)$.

In order to measure the performance of the classification process, $m$ metrics are used. In this paper we focus on metrics that are based on the unitary confusion matrix and, for the sake of much easier comparison, all these metrics are converted within the range $[-1, 1]$. Let us define $^j\gamma_i^k$ as the $j$-th of such metrics for the $c_i^k$ classifier operating on the $D_i^k$ dataset, where $j = 1, \ldots, m$. Since it is based on the unitary confusion matrix, $^j\gamma_i^k = {}^j\gamma_i^k \left( \lambda_i^k \right) = {}^j\gamma_i^k \left( \alpha_i^k, \beta_i^k, \delta_i^k \right)$.

Let us now define $\mu_j^k$ as the set of the $j$-th metric values corresponding to the $k$-th experiment $E^k = \left( C^k, D^k \right)$, that is, $\mu_j^k \equiv \left\{ {}^j\gamma_i^k \right\}$, $i = 1, 2, \cdots, n^k$. Additionally, the sets $\alpha^k \equiv \left\{ \alpha_i^k \right\}$, $\beta^k \equiv \left\{ \beta_i^k \right\}$ and $\delta^k \equiv \left\{ \delta_i^k \right\}$ are also defined.

## 2.2. Representation of Metrics

With these definitions, it is clear that the metric $\mu_j^k = \mu_j^k\left(\alpha^k, \beta^k, \delta^k\right)$ and hence it is a 4-dimensional function since $\mu_j^k$ (one dimension) depends on $\alpha^k, \beta^k$ *and* $\delta^k$ (three independent dimensions). To depict their values, a first approach could involve a 3D representation space where each $\left(\alpha_i^k, \beta_i^k, \delta_i^k\right)$ point is color-coded according to the value $^j\gamma_i^k\left(\alpha_i^k, \beta_i^k, \delta_i^k\right)$.

To show the different types of representations, let us define an arbitrary metric function

$$\mu_j^k\left(\alpha^k, \beta^k, \delta^k\right) = \frac{\sin\left(2\pi\alpha^k\delta^k\right) + \sin\left(2\pi\beta^k\right)}{2}. \tag{10}$$

This function is only used as an example, corresponds to no specific classification metric and has been selected for its aesthetic results. Figure 3 depicts the 3D representation for said example function. The $n^E = 1000$ pairs of classifiers and datasets used in the experiment $E^k = \left(C^k, D^k\right)$ are selected in such a way that the space $\left(\alpha^k, \beta^k, \delta^k\right)$ is covered with equally spaced points. The above figure may cause confusion, mainly when the number of points $(n^E)$ increases. An alternative is to slice the 3D graphic by a plane corresponding to a certain value of the imbalance coefficient. Figure 4a depicts such a slice in the 3D graphic for an arbitrary value $\delta = 0.75$ and Figure 4b shows the slice on a 2D plane.

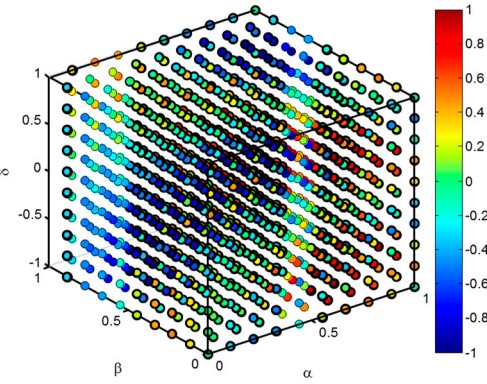

**Figure 3.** 3D representation of a 4-dimension metric value $\mu_j^k\left(\alpha^k, \beta^k, \delta^k\right)$. The value of the metric $\mu_j^k$ is colour-coded for every point in the $\left(\alpha^k, \beta^k, \delta^k\right)$ 3D space.

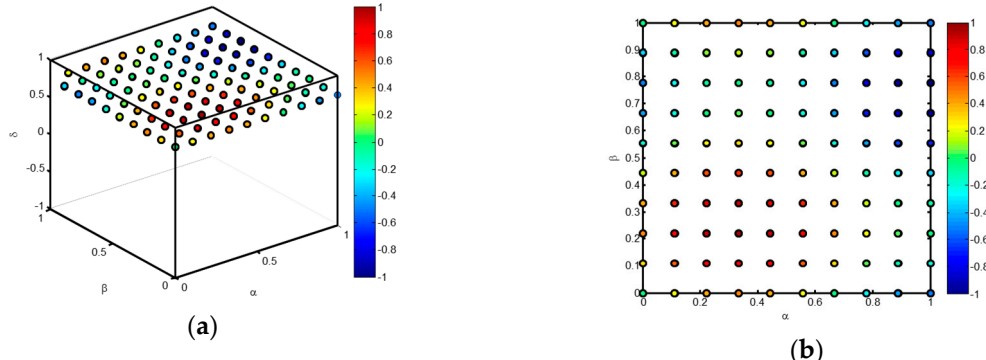

(a)

(b)

**Figure 4.** Representation of a metric value $\mu_j^k\left(\alpha^k, \beta^k\right)$ for $\delta = 0.75$. (a) Slice of the 3D graphic by a plane corresponding to $\delta = 0.75$; (b) 2D representation of the slice.

In the previous figure, the slice contains 100 values of the metric. However, to obtain a clearer understanding of the metric behaviour, a much larger number of points is recommended. For this purpose, the experiment is designed by selecting a set of virtual pairs of classifiers and datasets

$\left( C^k, D^k \right)$ in such a way that the plane $\left( \alpha^k, \beta^k \right)$ is fully covered. The result, as shown in Figure 5, appears as a heat map for a certain value of the imbalance coefficient ($\delta = 0.75$ in the example).

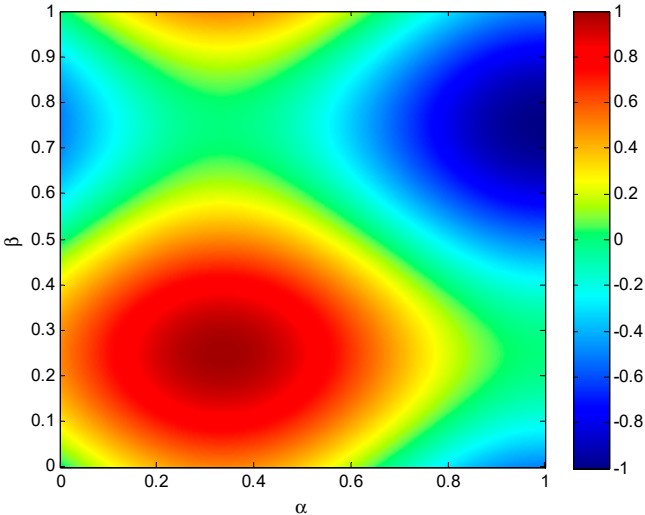

**Figure 5.** Heat map of a metric value $\mu_j^k \left( \alpha^k, \beta^k \right)$ for $\delta = 0.75$.

In order to analyse the behaviour of the metric for different values of the imbalance coefficient, a panel of heat maps can be used, as depicted in Figure 6.

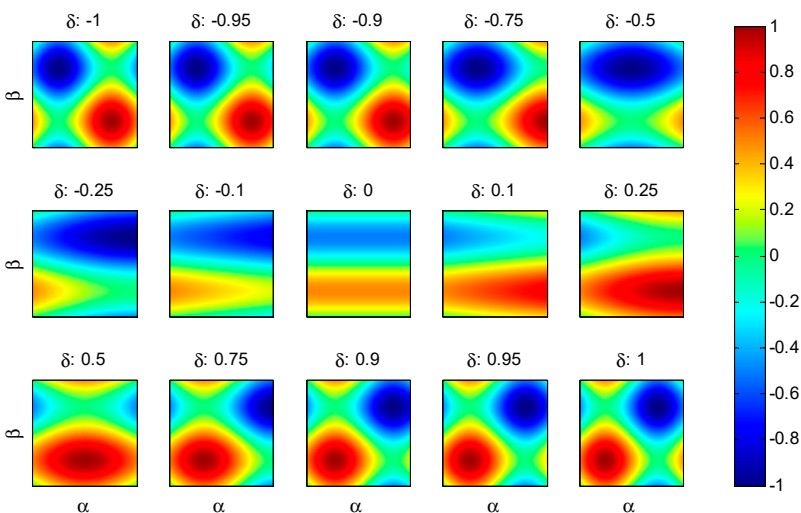

**Figure 6.** Panel of heat maps representing the metric $\mu_j^k \left( \alpha^k, \beta^k, \delta^k \right)$.

### 2.3. Transformations

The original baseline experiment $E^B = \left( C^B, D^B \right)$ is subject to various types of transformations. As a result of the $k$-th transformation, the metrics related to the baseline experiment $\mu_j^B \left( \alpha^B, \beta^B, \delta^B \right)$ are transformed into $\mu_j^k \left( \alpha^k, \beta^k, \delta^k \right)$, which can be written either as $\mu_j^k \left( \alpha^k, \beta^k, \delta^k \right) = T^k [\mu_j^B \left( \alpha^B, \beta^B, \delta^B \right)]$ or as

$$\mu_j^B \left( \alpha^B, \beta^B, \delta^B \right) \xrightarrow{T^k} \mu_j^k \left( \alpha^k, \beta^k, \delta^k \right). \tag{11}$$

It is said that the metric $\mu_j$ is symmetric under the transformation $T^k$ if $\mu_j^k = \mu_j^B$. Conversely, $\mu_j$ is called antisymmetric under $T^k$ (or symmetric under the complementary transformation $\overline{T}^k$) if $\mu_j^k = -\mu_j^B$. Analogously, it is said that the metrics $\mu_u$ and $\mu_v$ are cross-symmetric under the

transformation $T^k$ if $\mu_u^k = \mu_v^B$. Conversely, $\mu_u$ and $\mu_v$ are called anti-cross-symmetric under $T^k$ (or cross-symmetric under the complementary transformation $\overline{T}^k$) if $\mu_u^k = -\mu_v^B$.

### 2.3.1. One-Dimensional Transformations

One-dimensional transformations is the name given to those mirror reflections with respect to a single (one and only one) dimension of the 4-dimensional performance metric. Type $\alpha$ transformation implies that the $i$-th transformed classifier $\left(c_i^\alpha\right)$ shows a ratio of correctly classified positive elements $\left(\alpha_i^\alpha\right)$ which has the symmetric value of the ratio $\left(\alpha_i^B\right)$ obtained by the baseline classifier $\left(c_i^B\right)$. Since the values of such ratios lie within the range $[0, 1]$, the symmetry exists with respect to the hyperplane $\alpha = 0.5$ and can be stated as $\alpha_i^\alpha = 1 - \alpha_i^B$. An example of this transformation is depicted in Figure 7.

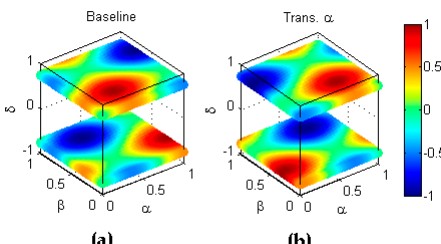

**Figure 7.** Transformation type $\alpha$ of a metric. (**a**) Baseline metric. (**b**) Reflection symmetry with respect to the hyperplane $\alpha = 0.5$.

Analogously, type $\beta$ transformation implies that the $i$-th transformed classifier $\left(c_i^\beta\right)$ shows a ratio of correctly classified negative elements $(\beta_i^\beta)$, which has the symmetric value of the ratio $(\beta_i^B)$ obtained by the baseline classifier $\left(c_i^B\right)$. Since the value of such ratios also lie within the range $[0, 1]$, the symmetry exists with respect to the hyperplane $\beta = 0.5$ and can be stated as $\beta_i^\beta = 1 - \beta_i^B$. An example of this transformation is depicted in Figure 8.

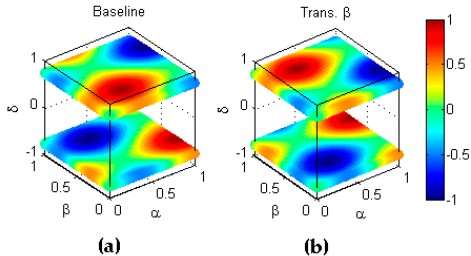

**Figure 8.** Transformation type $\beta$ of a metric. (**a**) Baseline metric; (**b**) Reflection symmetry with respect to the hyperplane $\beta = 0.5$.

Conversely, type $\delta$ transformation, which, instead of operating on classifiers, operates on datasets, implies that the $i$-th transformed dataset $\left(D_i^\delta\right)$ has an imbalance ratio $\left(\delta_i^\delta\right)$, which has the symmetric value of the imbalanced ratio $\left(\delta_i^B\right)$ in the baseline corresponding to dataset $\left(\delta_i^B\right)$. Since the value of such imbalance ratios lie within the range $[-1, 1]$, the symmetry exists with respect to the hyperplane $\delta = 0$ and can be stated as $\delta_i^\delta = -\delta_i^B$. An example of this transformation is depicted in Figure 9.

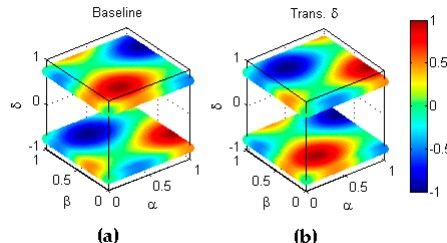

**Figure 9.** Transformation type $\delta$ of a metric. (**a**) Baseline metric. (**b**) Reflection symmetry with respect to the hyperplane $\delta = 0$.

Finally, type $\mu$ transformation jointly operates on classifiers and datasets in such a way that the $j$-th of performance metrics $^{j}\gamma_{i}^{\mu}$ for the $c_{i}^{\mu}$ classifier operating on the $D_{i}^{\mu}$ dataset has the symmetric value of the performance metric in the baseline experiment $\left(^{j}\gamma_{i}^{B}\right)$. Since the value of such metrics lie within the range $[-1, 1]$, the symmetry exists with respect to the hyperplane $\mu = 0$ and can be stated as $^{j}\gamma_{i}^{\mu} = -^{j}\gamma_{i}^{B}$. An example of this transformation is depicted in Figure 10 where it should be noted that the $\mu$ dimension is shown by the colour code of each point. Therefore, an inversion in $\mu$ is shown as a colour inversion.

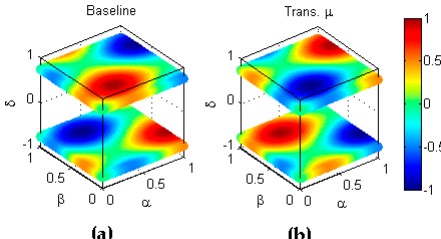

**Figure 10.** Transformation type $\mu$ of a metric. (**a**) Baseline metric. (**b**) Reflection symmetry with respect to the hyperplane $\mu = 0$.

### 2.3.2. Multidimensional Transformations

Let us now consider transformations that exchange two or more dimensions of the 4-dimensional performance metric. Firstly, let us define type $\sigma$ transformation as that which exchanges $\alpha$ and $\beta$ dimensions. This implies that the $i$-th transformed classifier/dataset pair $\left(c_{i}^{\sigma}, D_{i}^{\sigma}\right)$ shows a ratio of correctly classified positive elements $\left(\alpha_{i}^{\sigma}\right)$ which has the same value as the ratio of correctly classified negative elements $\left(\beta_{i}^{B}\right)$ obtained by the baseline classifier/dataset pair $\left(c_{i}^{B}, D_{i}^{B}\right)$. This exchange can be seen as the symmetry with respect to the hyperplane $\alpha = \beta$ (main diagonal of the $\alpha, \beta$ plane) and can be stated as $\alpha_{i}^{\sigma} = \beta_{i}^{B}$; $\beta_{i}^{\sigma} = \alpha_{i}^{B}$. An example of this transformation is depicted in Figure 11.

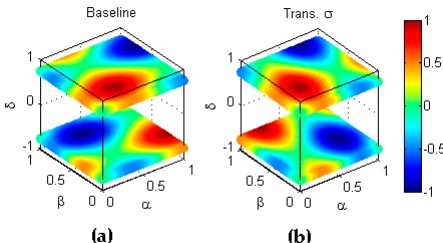

**Figure 11.** Transformation type $\sigma$ of a metric. (**a**) Baseline metric; (**b**) Reflection symmetry with respect to the hyperplane $\alpha = \beta$.

Although the four axes in these plots remain dimensionless, not all of them have the same meaning. So, $\alpha$ and $\beta$ are both ratios of correctly classified elements. It would be nonsensical, for instance, to rescale $\alpha$ without also rescaling $\beta$. However, $\delta$ has a completely different meaning

and its scale can and in fact does, differ from $\alpha$ and $\beta$. The same reasons can be applied to the axes $\mu$. Therefore, all the exchanges of multidimensional axes are meaningless, except the interchange of $\alpha$ and $\beta$. All the other remaining exchanges are dismissed in our study.

The one- and two-dimensional transformations described above are called basic transformations and are summarized in Table 1.

**Table 1.** Summary of basic transformations.

| Transformation | $\alpha^k$ | $\beta^k$ | $\delta^k$ | $\mu^k$ |
|---|---|---|---|---|
| $\alpha$ | $1 - \alpha^B$ | $\beta^B$ | $\delta^B$ | $\mu^B$ |
| $\beta$ | $\alpha^B$ | $1 - \beta^B$ | $\delta^B$ | $\mu^B$ |
| $\delta$ | $\alpha^B$ | $\beta^B$ | $-\delta^B$ | $\mu^B$ |
| $\mu$ | $\alpha^B$ | $\beta^B$ | $\delta^B$ | $-\mu^B$ |
| $\sigma$ | $\beta^B$ | $\alpha^B$ | $\delta^B$ | $\mu^B$ |

### 2.3.3. Combined Transformations.

More complex transformations can be obtained by concatenating basic transformations. For instance, applying basic transformation $\alpha$ ($T^\alpha$) and then basic transformation $\beta$ ($T^\beta$) produces a new combined transformation $T^{\alpha\beta} = T^\alpha \cdot T^\beta$ featured by $\alpha^{\alpha\beta} = 1 - \alpha^B$; $\beta^{\alpha\beta} = 1 - \beta^B$. As each of the one-dimensional transformations operates on an independent axis, they have the commutative and associative properties, that is, given 3 one-dimensional transformations, $T^U$, $T^V$ *and* $T^W$, it is true that $T^U \cdot T^V = T^V \cdot T^U$ and that that $(T^U \cdot T^V) \cdot T^W = T^U \cdot (T^V \cdot T^W)$.

However, bi-dimensional type $\sigma$ transformation $T^\sigma$ operates on the same axis as $T^\alpha$ and $T^\beta$. In this case, the order of transformation matters, as they do not have the commutative property. For instance, $T^{\alpha\sigma}[\mu_j^B] = T^\sigma \{ T^\alpha [\mu_j^B (\alpha^B, \beta^B, \delta^B)] \} = T^\sigma \{ \mu_j^B (1 - \alpha^B, \beta^B, \delta^B) \} = \mu_j^B (\beta^B, 1 - \alpha^B, \delta^B)$. On the other hand, $T^{\sigma\alpha}[\mu_j^B] = T^\alpha \{ T^\sigma [\mu_j^B (\alpha^B, \beta^B, \delta^B)] \} = T^\alpha \{ \mu_j^B (\beta^B, \alpha^B, \delta^B) \} = \mu_j^B (1\beta^B, \alpha^B, \delta^B)$. Therefore, it is clear that $T^{\alpha\sigma} \neq T^{\sigma\alpha}$.

Having 5 basic transformations and not initially considering their order, any combined transformation can be binary coded in terms of the presence/absence of each basic component. Therefore $2^5 = 32$ combinations are possible; only 31 if the identity transformation (coded 00000) is dismissed. In order to code a combined transformation, the order $\mu, \sigma, \delta, \beta, \alpha$ is used where transformation $\mu$ indicates the Most Significant Bit (MSB) and the transformation $\alpha$ specifies the Least Significant Bit (LSB). An example of this code is shown in Table 2. With this selection, codes greater than 15 contain a transformation type $\mu$, that is, they are useful in exploring antisymmetric behaviour. In the cases where the order of transformations matters, $\sigma = 1$ and ($\alpha = 1$ or $\beta = 1$), then their corresponding codes refer to various different combined transformations.

**Table 2.** Example of the coding of combined transformations.

| Transformation Code | $\mu$ | $\sigma$ | $\delta$ | $\beta$ | $\alpha$ |
|---|---|---|---|---|---|
| 28 | 1 | 1 | 1 | 0 | 0 |

A first example of combined transformations is that of the inverse labelling of classes. As stated above, the elements in every dataset belong to either of two classes, $G_1$ and $G_2$, which are called Positive ($P$) and Negative ($N$) classes, respectively. The inverse labelling transformation ($T^L$) explores the classification metric behaviour when the labelling of the classes is inverted, that is, when $G_2$ is called the Positive class and $G_1$ the Negative class. Let us consider the $i$-th classifier $c_i^L$ operating on its corresponding $D_i^L$ dataset. In the baseline experiment, the ratio of correctly classified positive elements ($\alpha_i^B$) refers to class $G_1$ *and* conversely ($\beta_i^B$) refers to class $G_2$. In the $T^L$ transformed experiment, the ratio of correctly classified positive elements ($\alpha_i^L$) refers to class $G_2$ *and* conversely ($\beta_i^L$) refers to

class $G_1$, which means that $\alpha_i^L = \beta_i^B$ and $\beta_i^L = \alpha_i^B$. That is, the first step of this transformation implies interchanging the axes $\alpha$ and $\beta$, which is equivalent to reflection symmetry with respect to the main diagonal, formerly defined as the basic transformation of type $\sigma$ (Figure 12b).

Additionally, in the baseline experiment, the number of positive elements ($P_i^B$) refers to class $G_1$, while in the $T^L$ transformed experiment, the number of positive elements ($P_i^L$) refers to class $G_2$, which means that $P_i^L = N_i^B$ and $N_i^L = P_i^B$, while the total number of elements remains unaltered: $M_i^L = M_i^B$. Therefore, by recalling Equation (5),

$$\delta_i^L \equiv 2\frac{P_i^L}{M_i^L} - 1 = 2\frac{N_i^B}{M_i^B} - 1 = 2\frac{M_i^B - P_i^B}{M_i^B} - 1 = -\left(2\frac{P_i^B}{M_i^B} - 1\right) = -\delta_i^B. \tag{12}$$

Hence, the second step of this transformation also implies reflection symmetry with respect to the hyperplane $\delta = 0$, previously defined as the basic transformation of type $\delta$ (Figure 12c).

Finally, the complementary transformation $\overline{T}^L$ involves a third and final step of inverting the sign of the metric, which is equivalent to reflection symmetry with respect to the hyperplane $\mu = 0$, formerly defined as the basic transformation of type $\mu$ (Figure 12d).

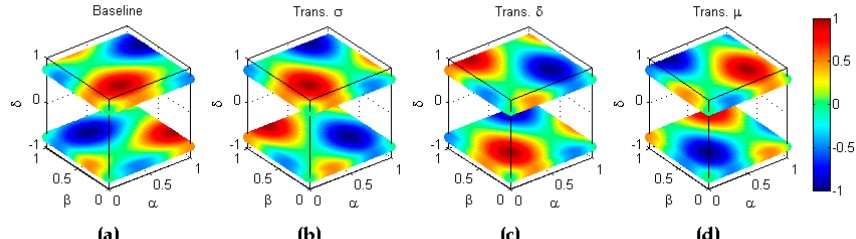

**Figure 12.** Transformation by inverse labelling of classes ($T^L$). (**a**) Baseline metric; (**b**) Reflection symmetry with respect to the main diagonal ($T^\sigma$); (**c**) Reflection symmetry with respect to the plane $\delta = 0$ ($T^\delta$); (**d**) Reflection symmetry with respect to the plane $\mu = 0$ (colour inversion, $T^\mu$).

Therefore, the inverse labelling transformation can be defined as $T^L = T^{\sigma\delta} = T^\sigma \cdot T^\delta$ and its complementary as $\overline{T}^L = T^{\sigma\delta\mu} = T^\sigma \cdot T^\delta \cdot T^\mu$, where

$$T^L : \mu_j^L\left(\alpha^L, \beta^L, \delta^L\right) = \mu_j^B\left(\beta^B, \alpha^B, -\delta^B\right). \tag{13}$$

A second example of combined transformations is given by the inverse-scoring transformation ($T^S$) which explores classification metric behaviour when the scoring of the classification results are inverted. In the baseline experiment, let us consider the $i$-th classifier $c_i^B$ operating on its corresponding $D_i^B$ dataset, thereby obtaining a ratio $\alpha_i^B$ of correctly classified positive elements and a ratio $\beta_i^B$ in the negative case. The $j$-th metric assigns a score of $^j\gamma_i^B(\alpha_i^B, \beta_i^B, \delta_i^B)$ to this result . High values of the score $^j\gamma_i^B$ usually correspond to high ratios $\alpha_i^B, \beta_i^B$. In the inverted score transformation ($T^S$), the $i$-th classifier $c_i^S$ operating on its corresponding $D_i^S$ dataset obtains a ratio $\alpha_i^S$ of correctly classified positive elements which is equal to the ratio of positive elements incorrectly classified in the baseline experiment, that is, $\alpha_i^S = 1 - \alpha_i^B$, which implies a type $\alpha$ transformation. Analogously, for the negative class, $\beta_i^S = 1 - \beta_i^B$, which implies a type $\beta$ transformation. If $\alpha_i^B, \beta_i^B$ have high values, then $\alpha_i^S$, $\beta_i^S$ will have low values and, to be consistent, the result should be marked with a low score. For that reason, the inverse scoring transform also implies a transformation type $\mu$, that is, it uses the symmetric value of the metric $^j\gamma_i^S = -^j\gamma_i^B$. Therefore, the inverse labelling transformation can be defined as $T^S = T^{\alpha\beta\mu} = T^\alpha \cdot T^\beta \cdot T^\mu$ where

$$T^S : \mu_j^S\left(\alpha^S, \beta^S, \delta^S\right) = \mu_j^B\left(1 - \alpha^B, 1 - \beta^B, \delta^B\right). \tag{14}$$

The results are depicted in Figure 13.

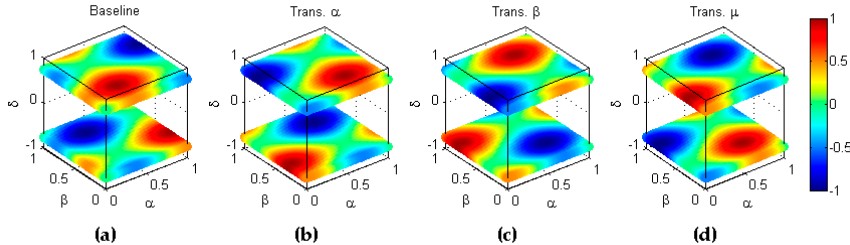

**Figure 13.** Transformation by inverse scoring ($T^S$). (**a**) Baseline metric; (**b**) Reflection symmetry with respect to the plane $\alpha = 0$ ($T^\alpha$); (**c**) Reflection symmetry with respect to the plane $\beta = 0$ ($T^\beta$); (**d**) Reflection symmetry with respect to the plane $\mu = 0$ (colour inversion, $T^\mu$).

A third example is that of the full inversion ($T^F$), which explores the classification metric behaviour when both the labelling ($T^L$) and the scores ($T^S$) are inverted. This transformation can be featured by the concatenation of their two components, which can be written as

$$T^F = T^L \cdot T^S = T^{\sigma\delta} \cdot T^{\alpha\beta\mu} = T^{\sigma\delta\alpha\beta\mu} = T^{\alpha\beta\delta\sigma\mu}. \tag{15}$$

$$T^F : \mu_j^F\left(\alpha^F, \beta^F, \delta^F\right) = -\mu_j^B\left(1 - \beta^B, 1 - \alpha^B, -\delta^B\right). \tag{16}$$

The results are depicted in Figure 14.

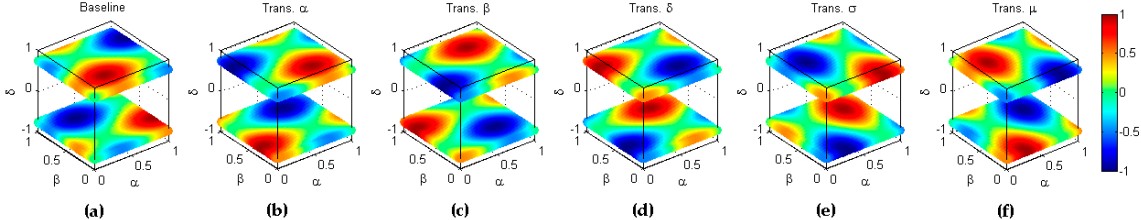

**Figure 14.** Transformation by full inversion scoring ($T^F$). (**a**) Baseline metric; (**b**) Reflection symmetry with respect to the plane $\alpha = 0$ ($T^\alpha$); (**c**) Reflection symmetry with respect to the plane $\beta = 0$ ($T^\beta$). (**c**) Reflection symmetry with respect to the main diagonal ($T^\sigma$); (**d**) Reflection symmetry with respect to the plane $\delta = 0$ ($T^\delta$). (**e**) Reflection symmetry with respect to the plane $\mu = 0$ (colour inversion, $T^\mu$).

Finally let us consider the $T^{\alpha\sigma\beta}$ transformation

$$T^{\alpha\sigma\beta}\left[\mu_j^B\left(\alpha^B, \beta^B, \delta^B\right)\right] = T^{\sigma\beta}\left[\mu_j^B\left(1 - \alpha^B, \beta^B, \delta^B\right)\right] = T^\beta\left[\mu_j^B\left(\beta^B, 1 - \alpha^B, \delta^B\right)\right]$$
$$= \mu_j^B\left(\beta^B, \alpha^B, \delta^B\right), \tag{17}$$

that is, $T^{\alpha\sigma\beta} = T^\sigma$. Analogously, it can be shown that $T^{\beta\sigma\alpha} = T^\sigma$.

## 2.4. Performance Metrics

Based on the binary confusion matrix, numerous performance metrics have been proposed [32–36]. For our study, the focus is placed on 10 of these metrics, which are summarized in Table 3. The terms used in that table are taken from the elements of a generic confusion matrix which can be stated as

$$cm = \begin{bmatrix} a & f \\ g & b \end{bmatrix}, \tag{18}$$

**Table 3.** Definition of classification performance metrics.

| Symbol | Metric | Scoring |
|--------|--------|---------|
| $SNS$ | Sensitivity | $\frac{a}{a+f}$ |
| $SPC$ | Specificity | $\frac{b}{b+g}$ |
| $PRC$ | Precision | $\frac{a}{a+g}$ |
| $NPV$ | Negative Predictive Value | $\frac{b}{b+f}$ |
| $ACC$ | Accuracy | $\frac{a+b}{a+f+b+g}$ |
| $F_1$ | $F_1$ score | $2\frac{PRC \cdot SNS}{PRC+SNS}$ |
| $GM$ | Geometric Mean | $\sqrt{SNS \cdot SPC}$ |
| $MCC$ | Matthews Correlation Coefficient | $\frac{a \cdot b - g \cdot f}{\sqrt{(a+g)(a+f)(b+g)(b+f)}}$ |
| $BM$ | Bookmaker Informedness | $SNS + SPC - 1$ |
| $MK$ | Markedness | $PPV + NPV - 1$ |

The last three metrics ($MCC$, $BM$ and $MK$) take values within the $[-1, 1]$ range, while the ranges for the first seven lie within the $[0, 1]$ interval. For comparison purposes, these metrics are used herein in their normalized version ($[-1, 1]$ interval). By naming a metric defined within the $[0, 1]$ interval as $\mu$, it can be normalized within the $[-1, 1]$ range by the expression

$$\mu_n \equiv 2\mu - 1. \tag{19}$$

It can easily be shown that all these metrics can be expressed as a function $\mu = \mu(\alpha, \beta, \delta)$.

Although only performance metrics based on the confusion matrix are considered, a marginal approach to Receiver Operating Characteristics (ROC) analysis [37] can also be carried out. In this analysis, the Area Under Curve ($AUC$) is commonly used as a performance metric. However, for classifiers offering only a label (and not a set of scores for each label) or when a single threshold is used on scores, the value of $AUCn$ and $BM$ are the same [38]. Therefore, in the forthcoming sections, whenever $BM$ is mentioned it could also be understood as $AUCn$.

*2.5. Exploring Symmetries*

In order to determine the existence of any symmetric or cross-symmetric behaviour on the 10 classification performance metrics described in the previous section, we should explore whether, for each metric (or pair of metrics), its baseline and any of the 31 combinations of transformations obtain the same result as that of the baseline of the same metric (symmetry) or any other metric (cross-symmetry). Moreover, many of these combined transformations must take the order into account. Therefore, several thousands of different analyses have to be undertaken. Although performing this task using analytical derivations is not an impossible assignment (preferably using some kind of symbolic computation), it is certainly arduous.

An alternative approach is to identify the distance of two metrics. More formally, for the $U$-th transformation, let us consider the $i$-th combination of classifier $c_i^U$ operating on the $D_i^U$ dataset. The classification result is measured using the $r$-th metric, $^r\gamma_i^U$. Similarly, for the $V$-th transform and the $i$-th combination of classifier $c_i^V$ operating on the $D_i^V$ dataset, let us measure its performance using the $s$-th metric, $^s\gamma_i^V$. The distance between these measures is defined as $dist\left(^r\gamma_i^U, {}^s\gamma_i^V\right) \equiv \left|^r\gamma_i^U - {}^s\gamma_i^V\right|$. The distance between the $r$-th metric $\mu_r^U = \{^r\gamma_i^U\}$ and the $s$-th metric $\mu_s^V = \{^s\gamma_i^V\}$ can then be defined as

$$dist\left(\mu_r^U, \mu_s^V\right) \equiv \frac{1}{n} \sum_{i=1}^{n} dist\left(^r\gamma_i^U, {}^s\gamma_i^V\right) = \frac{1}{n} \sum_{i=1}^{n} \left|^r\gamma_i^U, {}^s\gamma_i^V\right|. \tag{20}$$

Therefore, symmetric or cross-symmetric behaviour can be identified by a distance equal to zero.

It should be noted that if the $r$-th metric is symmetric under the $U$-th transformation, that is, $\mu_r^U = T^U(\mu_r^B) = \mu_r^B$ and also under the $V$-th transformations, $\mu_r^V = T^V(\mu_r^B) = \mu_r^B$, it will also be symmetric under the concatenation of the two transformations. In effect,

$$\mu_r^{UV} = T^V\left[T^U\left(\mu_r^B\right)\right] = T^V\left(\mu_r^B\right) = \mu_r^B. \tag{21}$$

Conversely, this is not true for cross-symmetries. If the $r$-th and $s$-th metric are cross-symmetric under the $U$-th transformation, that is, $\mu_r^U = T^U(\mu_r^B) = \mu_s^B$ and also under the $V$-th transformations, $\mu_r^V = T^V(\mu_r^B) = \mu_s^B$, they are not necessarily cross-symmetric under the concatenation of the two transformations. In effect,

$$\mu_r^{UV} = T^V\left[T^U\left(\mu_r^B\right)\right] = T^V\left(\mu_s^B\right) = \mu_r^B \neq \mu_s^B. \tag{22}$$

*2.6. Statistical Symmetries*

The symmetries of the performance metrics can also be explored from a statistical point of view. Let us recall that $D_i^k$ is the $i$-th dataset in the $k$-th experiment with an imbalance described by its imbalance coefficient $\delta_i^k$. The elements in $D_i^k$ are processed by the $c_i^k$ classifier in order to obtain a ratio of correctly classified positive $\alpha_i^k$ and negative $\beta_i^k$ elements. The $j$-th metric $^j\gamma_i^k$ is based on these values and hence $^j\gamma_i^k = {}^j\gamma_i^k\left(\alpha_i^k, \beta_i^k, \delta_i^k\right)$. Let us also recall that the set of all these values for $i = 1, \cdots, n^k$, are denoted $\mu_j^k = \left\{{}^j\gamma_i^k\right\}$, $\alpha^k = \left\{\alpha_i^k\right\}$, $\beta^k = \left\{\beta_i^k\right\}$ and $\delta^k = \left\{\delta_i^k\right\}$ and therefore $\mu_j^k = \mu_j^k\left(\alpha^k, \beta^k, \delta^k\right)$.

Let us now suppose that the elements $c_i^k, D_i^k$ in the experiments are randomly selected in such a way that $\alpha^k$, $\beta^k$ and $\delta^k$ are uniformly distributed within their respective ranges. Therefore, $\mu_j^k$ becomes a random variable, which can be statistically described.

First of all, the probability density function (pdf) of $\mu_j^k$: $pdf(\mu_j^k)$ is obtained and its symmetry (or lack thereof) is ascertained. A more precise assessment of the statistical symmetry can be obtained by computing the skewness, which is defined as

$$\xi_j^k \equiv skew\left(\mu_j^k\right) = E\left[\left(\frac{\mu_j^k - \overline{\mu}_j^k}{\sqrt{var\left(\mu_j^k\right)}}\right)^3\right], \tag{23}$$

where $\overline{\mu}_j^k$ is the mean of $\mu_j^k$ and $var\left(\mu_j^k\right)$ is its variance.

## 3. Results

*3.1. Identifying Symmetries*

The symmetric behaviour of the 10 metrics is first determined by means of computing the distance between the baseline and each of the 31 possible transformations, in accordance with Equation (20). The results are depicted in Figure 15. Each row shows the symmetries of a metric. In the columns are the 31 different transformations. Any given metric-transformation pair (small rectangles in the graphic) is shown in yellow if it has zero-distance with the metric baseline. The right-hand-side of the plot (whose code is greater than or equal to 16) corresponds to a combined transformation where the $\mu$ axis has been inverted, that is, where the transformation type $\mu$ is present. This is therefore the area for antisymmetric behaviour.

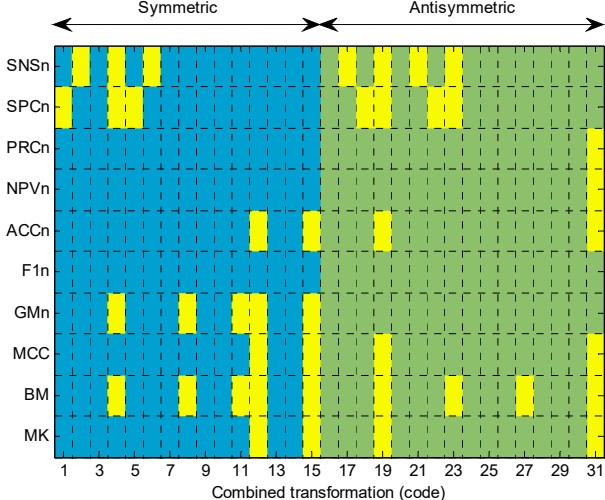

**Figure 15.** Symmetric behaviour of performance metrics for any combined transformation.

Let us first analyse each metric in terms of the accuracy (*ACCn*), the Matthews correlation coefficient (*MCC*) and the markedness (*MK*). These three metrics present a symmetric behaviour for the combined transformations shown in Table 4. For instance, the first row indicates that the three metrics are symmetric for a combination of the transformations $\delta$ and $\sigma$ taken in any order ($\delta\sigma$ or $\sigma\delta$), which corresponds to the code 12 (01100) for a coding scheme ($\mu, \sigma, \delta, \beta, \alpha$) where $\mu$ represents the Most Significant Bit and $\alpha$ represents the Least Significant Bit.

**Table 4.** Symmetric transformations of *ACCn*, *MCC* and *MK*.

| Code | $\mu$ | $\sigma$ | $\delta$ | $\beta$ | $\alpha$ | Specific Order | Any Order |
|------|-------|----------|----------|---------|----------|----------------|-----------|
| 12 | 0 | 1 | 1 | 0 | 0 | | $\delta\sigma$ |
| 15 | 0 | 1 | 1 | 1 | 1 | $\alpha\sigma\beta\ (=\sigma)$ <br> $\beta\sigma\alpha\ (=\sigma)$ | $\delta$ |
| 19 | 1 | 0 | 0 | 1 | 1 | | $\alpha\beta\mu$ |
| 31 | 1 | 1 | 1 | 1 | 1 | $\alpha\beta\sigma$ <br> $\beta\alpha\sigma$ <br> $\sigma\alpha\beta$ <br> $\sigma\beta\alpha$ | $\delta\mu$ |

The first case (code 12) corresponds to the transformation $T^{\sigma\delta}$, or, in other words, to the inverse labelling transformation $T^L = T^{\sigma\delta}$ which can be formulated for accuracy as

$$\mu_{ACCn}(\alpha, \beta, \delta) = \mu_{ACCn}(\beta, \alpha, -\delta). \tag{24}$$

The results are depicted in Figure 16.

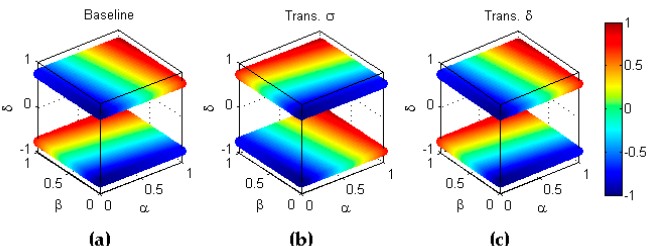

**Figure 16.** Symmetry of accuracy with respect to inverse labelling ($T^L$). (**a**) Baseline metric; (**b**) Reflection symmetry with respect to the main diagonal ($T^\sigma$); (**c**) Reflection symmetry with respect to the plane $\delta = 0$ ($T^\delta$).

The second case (code 15) corresponds to 4 transformations ordered in two different ways. In the first ordering, we have $T^{\alpha\sigma\beta\delta} = T^{\alpha\sigma\beta}\cdot T^{\delta}$. Recalling Equation (17), $T^{\alpha\sigma\beta} = T^{\sigma}$. It can therefore be written that $T^{\alpha\sigma\beta\delta} = T^{\sigma}\cdot T^{\delta} = T^{\sigma\delta} = T^{L}$, that is, it is equivalent to the inverse labelling transformation. The same result is obtained for $T^{\beta\sigma\alpha\delta}$. Hence, code 15 is the same case as code 12.

The third case (code 19) corresponds to the transformation $T^{\alpha\beta\mu}$, or, in other words, to the inverse scoring transformation $T^{S} = T^{\alpha\beta\mu}$, which can be formulated for accuracy as

$$\mu_{ACCn}(\alpha,\beta,\delta) = -\mu_{ACCn}(1-\alpha, 1-\beta, \delta). \tag{25}$$

The results are depicted in Figure 17.

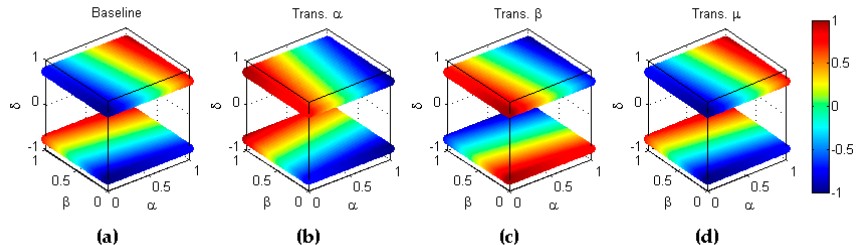

**Figure 17.** Symmetry of accuracy with respect to the inverse scoring ($T^{S}$). (**a**) Baseline metric; (**b**) Reflection symmetry with respect to the plane $\alpha = 0$ ($T^{\alpha}$); (**c**) Reflection symmetry with respect to the plane $\beta = 0$ ($T^{\beta}$); (**d**) Reflection symmetry with respect to the plane $\mu = 0$ (colour inversion, $T^{\mu}$).

Finally, code 31 corresponds to 5 transformations ordered in 4 different ways. In the first ordering we have $T^{\alpha\beta\sigma\delta\mu}$ but, by considering that the order of $T^{\delta}$ and $T^{\mu}$ are not relevant, it can also be written as $T^{\alpha\beta\sigma\delta\mu} = T^{\alpha\beta\delta\sigma\mu} = T^{\alpha\beta\delta}\cdot T^{\sigma\mu} = T^{L}\cdot T^{S} = T^{F}$, that is, it is equivalent to the full transformation. The same result is obtained for the 3 remaining orderings which can be formulated for accuracy as

$$\mu_{ACCn}(\alpha,\beta,\delta) = -\mu_{ACCn}(1-\beta, 1-\alpha, -\delta). \tag{26}$$

The results are depicted in Figure 18.

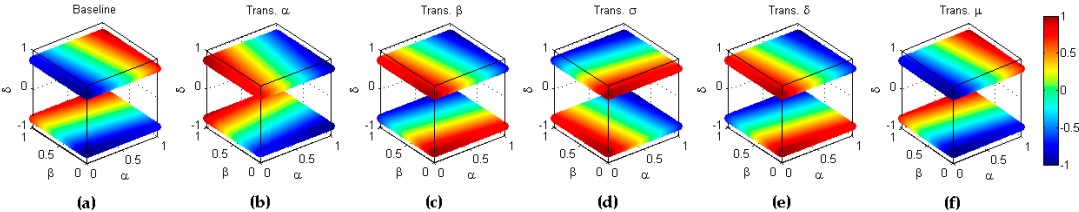

**Figure 18.** Symmetry of accuracy with respect to the full inversion ($T^{F}$). (**a**) Baseline metric; (**b**) Reflection symmetry with respect to the plane $\alpha = 0$ ($T^{\alpha}$); (**c**) Reflection symmetry with respect to the plane $\beta = 0$ ($T^{\beta}$); (**d**) Reflection symmetry with respect to the main diagonal ($T^{\sigma}$); (**e**) Reflection symmetry with respect to the plane $\delta = 0$ ($T^{\delta}$); (**f**) Reflection symmetry with respect to the plane $\mu = 0$ (colour inversion, $T^{\mu}$).

Let us now focus on precision (*PRCn*) and the negative prediction value (*NPVn*). These two metrics present a symmetric behaviour for the combined transformations shown in Table 5.

**Table 5.** Symmetric transformations of *PRCn* and *NPVn*.

| Code | $\mu$ | $\sigma$ | $\delta$ | $\beta$ | $\alpha$ | Specific Order | Any Order |
|------|-------|----------|----------|---------|----------|----------------|-----------|
| 31 | 1 | 1 | 1 | 1 | 1 | $\alpha\beta\sigma$<br>$\beta\alpha\sigma$<br>$\sigma\alpha\beta$<br>$\sigma\beta\alpha$ | $\delta\mu$ |

These two metrics present symmetric behaviour for only the combined transformations code 31 (11111) which, in any of its ordering, is equivalent to the full inversion $T^F = T^L \cdot T^S = T^{\alpha\beta\delta\sigma\mu}$ and can be formulated for precision as

$$\mu_{PRCn}(\alpha, \beta, \delta) = -\mu_{PRCn}(1 - \beta, 1 - \alpha, -\delta). \tag{27}$$

In other words, precision is symmetric with respect to the concatenation of inverse labelling and the inverse scoring transformations. The results are depicted in Figure 19.

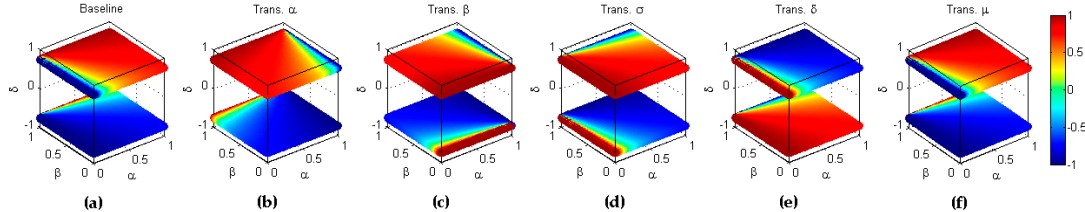

**(a)**      **(b)**      **(c)**      **(d)**      **(e)**      **(f)**

**Figure 19.** Symmetry of precision with respect to the full inversion ($T^F$). (**a**) Baseline metric; (**b**) Reflection symmetry with respect to the plane $\alpha = 0$ ($T^\alpha$); (**c**) Reflection symmetry with respect to the plane $\beta = 0$ ($T^\beta$); (**d**) Reflection symmetry with respect to the main diagonal ($T^\sigma$); (**e**) Reflection symmetry with respect to the plane $\delta = 0$ ($T^\delta$); (**f**) Reflection symmetry with respect to the plane $\mu = 0$ (colour inversion, $T^\mu$).

Let us now analyse the geometric mean (*GMn*), which presents symmetric behaviour for the combined transformations shown in Table 6.

**Table 6.** Symmetric transformations of *GMn*.

| Code | $\mu$ | $\sigma$ | $\delta$ | $\beta$ | $\alpha$ | Specific Order | Any Order |
|------|-------|----------|----------|---------|----------|----------------|-----------|
| 4 | 0 | 0 | 1 | 0 | 0 | | $\delta$ |
| 8 | 0 | 1 | 0 | 0 | 0 | | $\sigma$ |
| 11 | 0 | 1 | 0 | 1 | 1 | $\alpha\sigma\beta \, (= \sigma)$ <br> $\beta\sigma\alpha \, (= \sigma)$ | |
| 12 | 0 | 1 | 1 | 0 | 0 | | $\delta\sigma$ |
| 15 | 0 | 1 | 1 | 1 | 1 | $\alpha\sigma\beta \, (= \sigma)$ <br> $\beta\sigma\alpha \, (= \sigma)$ | $\delta$ |

In first place, code 4 corresponds to $T^\delta$. In fact, this metric is not only symmetric with respect to $\delta$ but also independent of $\delta$, as it can be seen in Table 3. Secondly, combined transformations coded as 8 and 11 are equivalent to the $T^\sigma$ transformation, that is, *GMn* is symmetric with respect to the diagonal in the $\alpha, \beta$ plane. This can be formulated as

$$\mu_{GMn}(\alpha, \beta) = \mu_{GMn}(\beta, \alpha). \tag{28}$$

Finally, codes 12 and 15 imply concatenating $T^\delta$ to $T^\sigma$ but as the metric is independent of $\delta$, it is again equivalent to $T^\sigma$, that is, $T^{\sigma\delta} = T^\sigma \cdot T^\delta = T^\sigma$. These results are depicted in Figure 20.

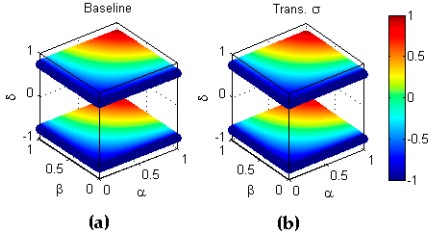

**(a)**      **(b)**

**Figure 20.** Symmetry of geometric mean with respect to $T^\sigma$. (**a**) Baseline metric; (**b**) Reflection symmetry with respect to the main diagonal ($T^\sigma$).

In the case of bookmaker informedness (*BM*), the symmetric behaviour is obtained for the combined transformations shown in Table 7.

**Table 7.** Symmetric transformations of *BM*.

| Code | $\mu$ | $\sigma$ | $\delta$ | $\beta$ | $\alpha$ | Specific Order | Any Order |
|------|-------|----------|----------|---------|----------|----------------|-----------|
| 4 | 0 | 0 | 1 | 0 | 0 | | $\delta$ |
| 8 | 0 | 1 | 0 | 0 | 0 | | $\sigma$ |
| 11 | 0 | 1 | 0 | 1 | 1 | $\alpha\sigma\beta\ (=\sigma)$ <br> $\beta\sigma\alpha\ (=\sigma)$ | |
| 12 | 0 | 1 | 1 | 0 | 0 | | $\delta\sigma$ |
| 15 | 0 | 1 | 1 | 1 | 1 | $\alpha\sigma\beta\ (=\sigma)$ <br> $\beta\sigma\alpha\ (=\sigma)$ | $\delta$ |
| 19 | 1 | 0 | 0 | 1 | 1 | | $\alpha\beta\mu$ |
| 23 | 1 | 0 | 1 | 1 | 1 | | $\alpha\beta\delta\mu$ |
| 27 | 1 | 1 | 0 | 1 | 1 | $\alpha\beta\sigma$ <br> $\beta\alpha\sigma$ <br> $\sigma\alpha\beta$ <br> $\sigma\beta\alpha$ | $\mu$ |
| 31 | 1 | 1 | 1 | 1 | 1 | $\alpha\beta\sigma$ <br> $\beta\alpha\sigma$ <br> $\sigma\alpha\beta$ <br> $\sigma\beta\alpha$ | $\delta\mu$ |

Again code 4 corresponds to $T^\delta$ as a consequence that this metric is independent of $\delta$ (see Table 3). Secondly, combined transformations coded as 8 and 11 are equivalent to the $T^\sigma$ transformation, that is, *BM* is symmetric with respect to the diagonal in the $\alpha, \beta$ plane. This can be formulated as

$$\mu_{BM}(\alpha, \beta) = \mu_{BM}(\beta, \alpha). \tag{29}$$

Additionally, codes 12 and 15 imply concatenating $T^\delta$ to $T^\sigma$ but since the metric is independent of $\delta$, it is again equivalent to $T^\sigma$, that is, $T^{\sigma\delta} = T^\sigma \cdot T^\delta = T^\sigma$. These results are depicted in Figure 21.

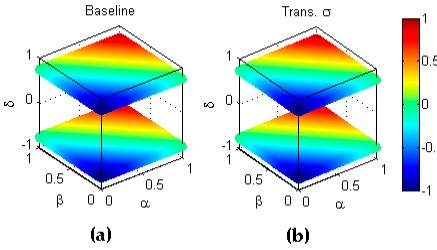

**Figure 21.** Symmetry of bookmaker informedness with respect to $T^\sigma$. (**a**) Baseline metric; (**b**) Reflection symmetry with respect to the main diagonal ($T^\sigma$).

Code 19 and also code 23 since the metric does not depend on $\delta$, correspond to the transformation $T^{\alpha\beta\mu}$ or, in other words, to the inverse scoring transformation $T^S = T^{\alpha\beta\mu}$, which can be formulated for bookmaker informedness as

$$\mu_{BM}(\alpha, \beta) = -\mu_{BM}(1 - \alpha, 1 - \beta) = -\mu_{BM}(1 - \beta, 1 - \alpha). \tag{30}$$

The results are depicted in Figure 22.

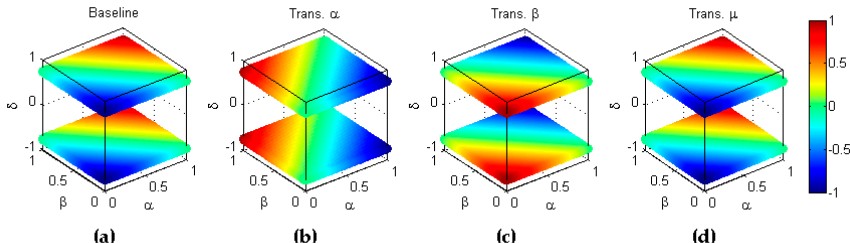

**Figure 22.** Symmetry of bookmaker informedness with respect to the inverse scoring ($T^S$). (**a**) Baseline metric; (**b**) Reflection symmetry with respect to the plane $\alpha = 0$ ($T^\alpha$); (**c**) Reflection symmetry with respect to the plane $\beta = 0$ ($T^\beta$); (**d**) Reflection symmetry with respect to the plane $\mu = 0$ (colour inversion, $T^\mu$).

In other words, the bookmaker informedness is symmetric with respect to the inverse labelling and to the inverse scoring transformations. This implies that it is also symmetric with respect to the concatenations of these two transforms, which occurs in codes 27 and 31 (recall that the latter is independent of $\delta$) corresponding to the full inversion $T^F = T^L + T^S = T^{\alpha\beta\delta\sigma\mu}$, which can be formulated as

$$\mu_{BM}(\alpha, \beta) = -\mu_{BM}(1 - \beta, 1 - \alpha). \tag{31}$$

The results are depicted in Figure 23.

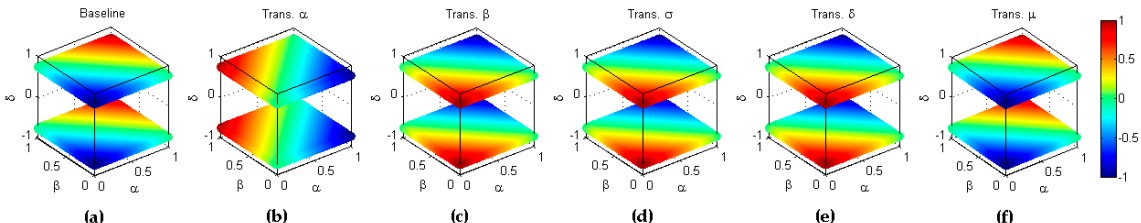

**Figure 23.** Symmetry of bookmaker informedness with respect to the full inversion ($T^F$). (**a**) Baseline metric; (**b**) Reflection symmetry with respect to the plane $\alpha = 0$ ($T^\alpha$); (**c**) Reflection symmetry with respect to the plane $\beta = 0$ ($T^\beta$); (**c**) Reflection symmetry with respect to the main diagonal ($T^\sigma$); (**d**) Reflection symmetry with respect to the plane $\delta = 0$ ($T^\delta$); (**e**) Reflection symmetry with respect to the plane $\mu = 0$ (colour inversion, $T^\mu$).

In the case of sensitivity ($SNSn$), the symmetric behaviour is found for the combined transformations shown in Table 8.

**Table 8.** Symmetric transformations of sensitivity.

| Code | $\mu$ | $\sigma$ | $\delta$ | $\beta$ | $\alpha$ | Specific Order | Any Order |
|------|-------|----------|----------|---------|----------|----------------|-----------|
| 2    | 0     | 0        | 0        | 1       | 0        |                | $\beta$   |
| 4    | 0     | 0        | 1        | 0       | 0        |                | $\delta$  |
| 6    | 0     | 0        | 1        | 1       | 0        |                | $\beta\delta$ |
| 17   | 1     | 0        | 0        | 0       | 1        |                | $\alpha\mu$ |
| 19   | 1     | 0        | 0        | 1       | 1        |                | $\alpha\beta\mu$ |
| 21   | 1     | 0        | 1        | 0       | 1        |                | $\alpha\delta\mu$ |
| 23   | 1     | 0        | 1        | 1       | 1        |                | $\alpha\beta\delta\mu$ |

Codes 2 and 4 correspond to $T^\beta$ and $T^\delta$ as a consequence of this metric being independent of $\beta$ and $\delta$ (see Table 3). Code 19 (and also codes 17, 21 and 23 since the metric does not depend on $\beta$ nor

$\delta$) corresponds to the transformation $T^{\alpha\beta\mu}$, or, in other words, to the inverse scoring transformation $T^S = T^{\alpha\beta\mu}$, which can be formulated as

$$\mu_{SNSn}(\alpha) = -\mu_{SNSn}(1-\alpha).\tag{32}$$

This result is depicted in Figure 24.

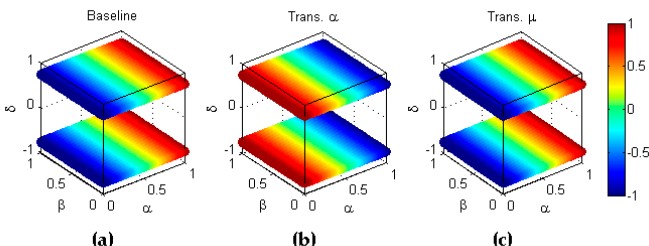

**Figure 24.** Symmetry of sensitivity with respect to the combined transformation ($T^{\alpha\mu}$). (**a**) Baseline metric; (**b**) Reflection symmetry with respect to the plane $\alpha = 0$ ($T^{\alpha}$); (**c**) Reflection symmetry with respect to the plane $\mu = 0$ (colour inversion, $T^{\mu}$).

On considering the specificity ($SPCn$), its symmetric behaviour is shown in Table 9.

**Table 9.** Symmetric transformations of specificity.

| Code | $\mu$ | $\sigma$ | $\delta$ | $\beta$ | $\alpha$ | Specific Order | Any Order |
|------|-------|----------|----------|---------|----------|----------------|-----------|
| 1    | 0     | 0        | 0        | 0       | 1        |                | $\alpha$  |
| 4    | 0     | 0        | 1        | 0       | 0        |                | $\delta$  |
| 5    | 0     | 0        | 1        | 0       | 1        |                | $\alpha\delta$ |
| 18   | 1     | 0        | 0        | 1       | 0        |                | $\beta\mu$ |
| 19   | 1     | 0        | 0        | 1       | 1        |                | $\alpha\beta\mu$ |
| 22   | 1     | 0        | 1        | 1       | 0        |                | $\beta\delta\mu$ |
| 23   | 1     | 0        | 1        | 1       | 1        |                | $\alpha\beta\delta\mu$ |

Codes 1 and 4 corresponds to $T^{\alpha}$ and $T^{\delta}$ as a consequence of this metric being independent of $\alpha$ and $\delta$ (see Table 3). Code 19 (and also codes 18, 22 and 23 as the metric depends neither on $\alpha$ nor on $\delta$) corresponds to the transformation $T^{\alpha\beta\mu}$, that is, to the inverse scoring transformation $T^S = T^{\alpha\beta\mu}$, which can be formulated as

$$\mu_{SPCn}(\beta) = -\mu_{SPCn}(1-\beta).\tag{33}$$

This result is depicted in Figure 25.

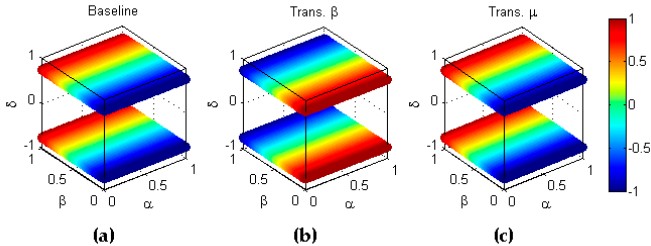

**Figure 25.** Symmetry of specificity with respect to the combined transformation ($T^{\beta\mu}$) (**a**) Baseline metric; (**b**) Reflection symmetry with respect to the plane $\beta = 0$ ($T^{\beta}$); (**c**) Reflection symmetry with respect to the plane $\mu = 0$ (colour inversion, $T^{\mu}$).

Finally, it can be observed that the $F_1n$ score metric is not symmetric under any transformation. The results for each metric are summarized in Table 10.

**Table 10.** Summary of symmetries.

| Metric | Independent of | | | Symmetry (under Inversion of) | | |
|---|---|---|---|---|---|---|
| | $\alpha$ | $\beta$ | $\delta$ | Labelling | Scoring | Full |
| *SNSn* | | ✓ | ✓ | | ✓ | |
| *SPCn* | ✓ | | ✓ | | ✓ | |
| *PRCn* | | | | | | ✓ |
| *NPVn* | | | | | | ✓ |
| *ACCn* | | | | ✓ | ✓ | ✓ |
| $F_1n$ | | | | | | |
| *GMn* | | ✓ | | ✓ | | |
| *MCC* | | | | ✓ | ✓ | ✓ |
| *BM* | | ✓ | | ✓ | ✓ | ✓ |
| *MK* | | | | ✓ | ✓ | ✓ |

### 3.2. Identifying Cross-Symmetries

In order to explore whether any cross-symmetry can be identified among the 10 metrics, we have computed the distance (using Equation (20)) of the baseline of each metric (and its 31 possible transformations), to the remaining baseline metrics. The results are depicted in Figure 26. Each row corresponds to the baseline of a metric and each column to the baseline and its 31 transformations of the other metric. Any given metric-metric pair (small squares in the graphic) is shown in yellow if it has zero-distance for any possible transformation.

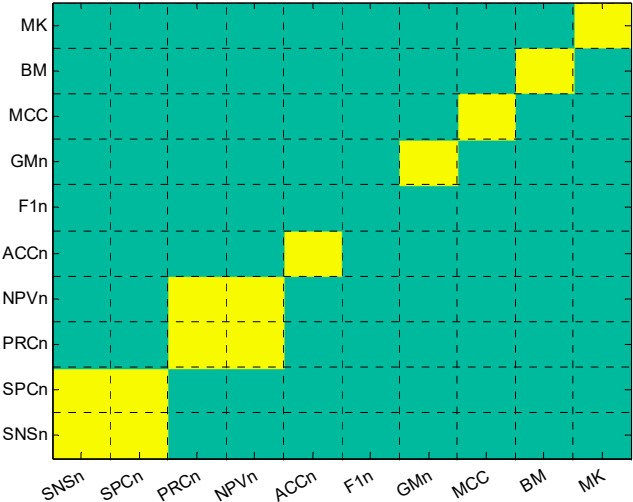

**Figure 26.** Cross-symmetric behaviour of performance metrics for any combined transformation.

The diagonal presents a summary of the results explored in the previous section, that is, every metric, except for the $F_1n$ score, presents some kind of symmetry under some transformation. The cases of cross-symmetries appear in the elements off diagonal. Two cross-symmetries arise: the $SNSn - SPCn$ and the $PRCn - NPVn$.

In order to attain a deeper insight into these cross-symmetries, let us consider, for each of the two pairs, the distances between the baseline of the first metric in the pair and the full set of transformations (including the baseline) of the second metric. The results are depicted in Figure 27. Each row shows the

cross-symmetries of a pair of metrics. In the columns are the 32 different transformations (including the baseline) of the second metric in the pair. Any given (second-metric transformation) pair (small squares in the graphic) is shown in yellow if it has zero-distance with the first metric baseline. As in Figure 15, the right-hand-side of the plot (with code greater than or equal to 16) corresponds to combined transformation where the $\mu$ axis has been inverted, that is, where the transformation type $\mu$ is present. This is therefore the area for antisymmetric behaviour.

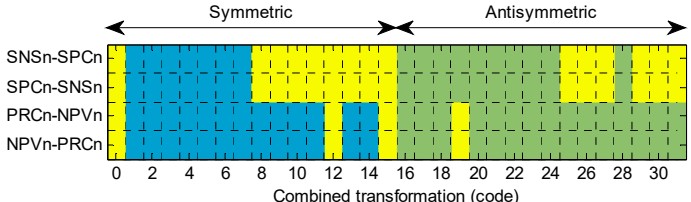

**Figure 27.** Cross-symmetric behaviour for any combined transformation.

Let us first analyse each pair of metrics in terms of the $PRCn - NPVn$ or $PRCn - NPVn$ pair, which present cross-symmetric behaviour for the combined transformations shown in Table 11.

**Table 11.** Cross-symmetric transformations of the $PRCn - NPVn$ pair.

| Code | $\mu$ | $\sigma$ | $\delta$ | $\beta$ | $\alpha$ | Specific Order | Any Order |
|------|-------|----------|----------|---------|----------|----------------|-----------|
| 12 | 0 | 1 | 1 | 0 | 0 | | $\delta\sigma$ |
| 15 | 0 | 1 | 1 | 1 | 1 | $\alpha\sigma\beta \, (=\sigma)$ <br> $\beta\sigma\alpha \, (=\sigma)$ | $\delta$ |
| 19 | 1 | 0 | 0 | 1 | 1 | | $\alpha\beta\mu$ |

Codes 12 and 15 correspond to the transformation $T^{\sigma\delta}$ or, in other words, to the inverse labelling transformation $T^L = T^{\sigma\delta}$, which can be formulated as

$$\mu_{PRCn}(\alpha, \beta, \delta) = \mu_{NPVn}(\beta, \alpha, -\delta). \tag{34}$$

The results are depicted in Figure 28.

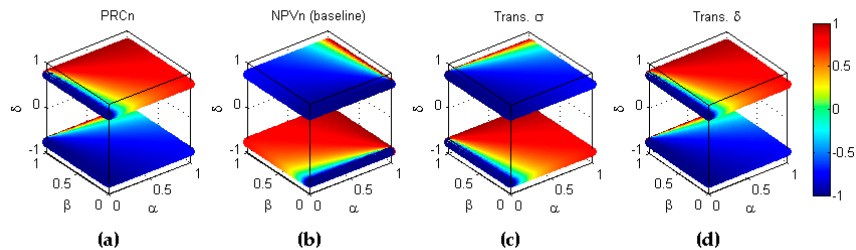

**Figure 28.** Cross-symmetry of the $PRCn - NPVn$ pair with respect to the inverse labelling ($T^L$). (**a**) Baseline $PRCn$ metric; (**b**) Baseline $NPVn$ metric; (**c**) Reflection symmetry of $NPVn$ with respect to the main diagonal ($T^\sigma$); (**d**) Reflection symmetry of $NPVn$ with respect to the plane $\delta = 0$ ($T^\delta$).

Code 19 corresponds to the transformation $T^{\alpha\beta\mu}$ or, in other words, to the inverse scoring transformation $T^S = T^{\alpha\beta\mu}$, which can be formulated as

$$\mu_{PRCn}(\alpha, \beta, \delta) = -\mu_{NPVn}(1 - \alpha, 1 - \beta, \delta). \tag{35}$$

The results are depicted in Figure 29.

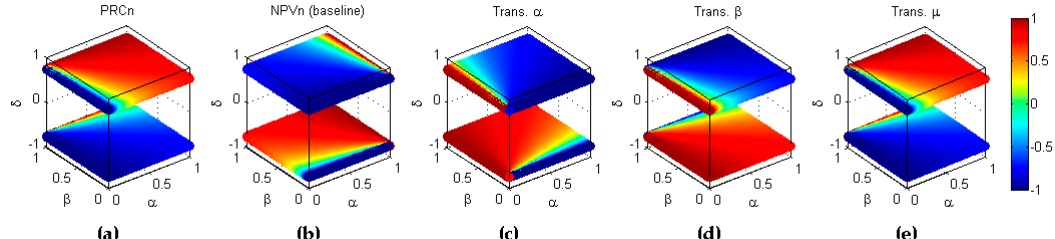

**Figure 29.** Cross-symmetry of the $PRCn - NPVn$ pair with respect to the inverse scoring ($T^S$). (**a**) Baseline $PRCn$ metric; (**b**) Baseline $NPVn$ metric. (**c**) Reflection symmetry of $NPVn$ with respect to the plane $\alpha = 0$ ($T^\alpha$); (**d**) Reflection symmetry of $NPVn$ with respect to the plane $\beta = 0$ ($T^\beta$); (**e**) Reflection symmetry of $NPVn$ with respect to the plane $\mu = 0$ (colour inversion, $T^\mu$).

Although the $PRCn - NPVn$ pair is cross-symmetric with respect to the inverse labelling and to the inverse scoring transformations, this does not imply that it is also cross-symmetric with respect to the concatenations of these two transforms (see Equation (22)). This is the reason why code 31 (corresponding to the full inversion $T^F = T^L + T^S = T^{\alpha\beta\delta\sigma\mu}$ is not present in Table 11.

The results for the pair $NPVn - PRCn$ are exactly the same. Therefore,

$$\mu_{NPVn}(\alpha, \beta, \delta) = \mu_{PRCn}(\beta, \alpha, -\delta) = -\mu_{PRCn}(1 - \alpha, 1 - \beta, \delta). \tag{36}$$

Let us now consider the pair of metrics $SNSn - SPCn$ and its cross-symmetric behaviour, which is found for the combined transformations shown in Table 12.

**Table 12.** Cross-symmetric transformations of the $SNSn - SPCn$ pair.

| Code | $\mu$ | $\sigma$ | $\delta$ | $\beta$ | $\alpha$ | Specific Order | Any Order |
|------|-------|----------|----------|---------|----------|----------------|-----------|
| 8 | 0 | 1 | 0 | 0 | 0 | | $\sigma$ |
| 9 | 0 | 1 | 0 | 0 | 1 | $\alpha\sigma$ | |
| 10 | 0 | 1 | 0 | 1 | 0 | $\sigma\beta$ | |
| 11 | 0 | 1 | 0 | 1 | 1 | $\alpha\sigma\beta\ (=\sigma)$ $\sigma\beta\alpha\ (=\sigma)$ | |
| 12 | 0 | 1 | 1 | 0 | 0 | | $\sigma\delta$ |
| 13 | 0 | 1 | 1 | 0 | 1 | $\alpha\sigma$ | $\delta$ |
| 14 | 0 | 1 | 1 | 1 | 0 | $\sigma\beta$ | $\delta$ |
| 15 | 0 | 1 | 1 | 1 | 1 | $\alpha\sigma\beta\ (=\sigma)$ $\sigma\beta\alpha\ (=\sigma)$ | $\delta$ |
| 25 | 1 | 1 | 0 | 0 | 1 | $\sigma\alpha$ | $\mu$ |
| 26 | 1 | 1 | 0 | 1 | 0 | $\beta\sigma$ | $\mu$ |
| 27 | 1 | 1 | 0 | 1 | 1 | $\alpha\beta\sigma$ $\beta\alpha\sigma$ $\sigma\alpha\beta$ $\sigma\beta\alpha$ | $\mu$ |
| 29 | 1 | 1 | 1 | 0 | 1 | $\sigma\alpha$ | $\delta\mu$ |
| 30 | 1 | 1 | 1 | 1 | 0 | $\beta\sigma$ | $\delta\mu$ |
| 31 | 1 | 1 | 1 | 1 | 1 | $\alpha\beta\sigma$ $\beta\alpha\sigma$ $\sigma\alpha\beta$ $\sigma\beta\alpha$ | $\delta\mu$ |

Since specificity remains independent from $\delta$ (see Table 3), codes 25, 11, 12 and 15 correspond to $T^{\sigma\delta}$, that is, to the inverse labelling which can be formulated as

$$\mu_{SNSn}(\alpha, \beta) = \mu_{SPCn}(\beta, \alpha). \tag{37}$$

Additionally, since specificity is also independent of $\alpha$, then codes 9 ($T^{\alpha\sigma}$) and 13 ($T^{\alpha\sigma\delta}$) are equivalent to $T^{\sigma\delta}$. Moreover, after a $T^{\sigma}$ transformation, the resulting metric has no dependence on $\beta$ (due to the axis inversion) and hence codes 10 ($T^{\sigma\beta}$) and 14 ($T^{\sigma\beta\delta}$) are also equivalent to $T^{\sigma\delta}$. These results are depicted in Figure 30.

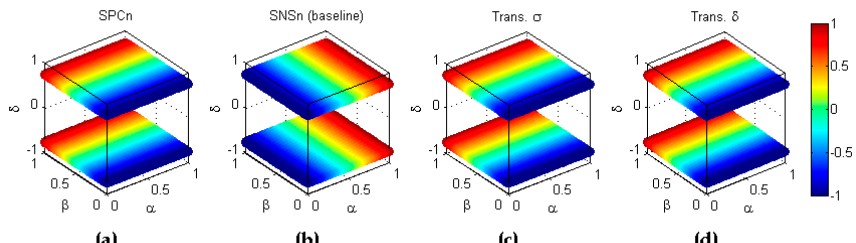

**Figure 30.** Cross-symmetry of the $SNSn - SPCn$ pair with respect to the inverse labelling ($T^L$). (**a**) Baseline $SNSn$ metric; (**b**) Baseline $SPCn$ metric; (**c**) Reflection symmetry of $SPCn$ with respect to the main diagonal ($T^{\sigma}$); (**d**) Reflection symmetry of $SPCn$ with respect to the plane $\delta = 0$ ($T^{\delta}$).

On the other hand, code 31 corresponds to full inversion transformation $T^F = T^{\sigma\delta\alpha\beta\mu}$, which can be formulated as

$$\mu_{SNSn}(\alpha, \beta) = -\mu_{SPCn}(1 - \beta, 1 - \alpha). \tag{38}$$

It can be shown that the remaining codes (25, 26, 27, 29 and 30) are also equivalent to $T^F$. Moreover, after a $T^{\sigma}$ transformation, the resulting metric does not depend on $\beta$ (due to the axis inversion) and hence codes 10 ($T^{\sigma\beta}$) and 14 ($T^{\sigma\beta\delta}$) are also equivalent to $T^{\sigma\delta}$. These results are depicted in Figure 31.

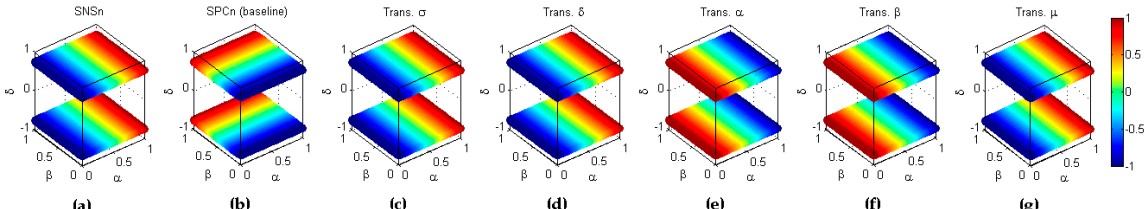

**Figure 31.** Cross-symmetry of the $SNSn - SPCn$ pair with respect to the full inversion ($T^F$). (**a**) Baseline $SNSn$ metric. (**b**) Baseline $SPCn$ metric. (**c**) Reflection symmetry of $SPCn$ with respect to the main diagonal ($T^{\sigma}$). (**d**) Reflection symmetry of $SPCn$ with respect to the plane $\delta = 0$ ($T^{\delta}$). (**e**) Reflection symmetry with respect to the plane $\alpha = 0$ ($T^{\alpha}$). (**f**) Reflection symmetry with respect to the plane $\beta = 0$ ($T^{\beta}$). (**g**) Reflection symmetry with respect to the plane $\mu = 0$ (colour inversion, $T^{\mu}$).

The results for the pair $SPCn - SNSn$ are exactly the same, so

$$\mu_{SPCn}(\alpha, \beta) = \mu_{SNSn}(\beta, \alpha) = -\mu_{SNSn}(1 - \beta, 1 - \alpha). \tag{39}$$

The results for every pair of cross-symmetric metrics are summarized in Table 13.

**Table 13.** Summary of cross-symmetries.

| Metric | Cross-Symmetry (under Inversion of) | | |
|--------|-------------|---------|------|
| | Labelling | Scoring | Full |
| *SNSn* | *SPCn* | *(SNSn)* | *SPCn* |
| *SPCn* | *SNSn* | *(SPCn)* | *SNSn* |
| *PRCn* | *NPVn* | *NPVn* | *(PRCn)* |
| *NPVn* | *PRCn* | *PRCn* | *(NPVn)* |

*3.3. Skewness of the Statistical Descriptions of the Metrics*

In order to explore the symmetric behaviour of the statistical descriptions of the metrics, let us recall that, for the baseline experiment, $\mu_j^B = \mu_j^B(\alpha^B, \beta^B, \delta^B)$ can be considered a statistical variable. First of all, let us select a subset of the $\mu_j^B$ corresponding to a certain value $\delta_0$ of the imbalance coefficient, that is, $\mu_j^B(\alpha^B, \beta^B, \delta_0)$ and obtain its probability density function (pdf) which will be called local pdf (since it is obtained solely for a value of $\delta^B$). The results $pdf(\mu_j^k, \delta_0)$ for every metric with $\delta^B = 0.5$ are shown in Figure 32.

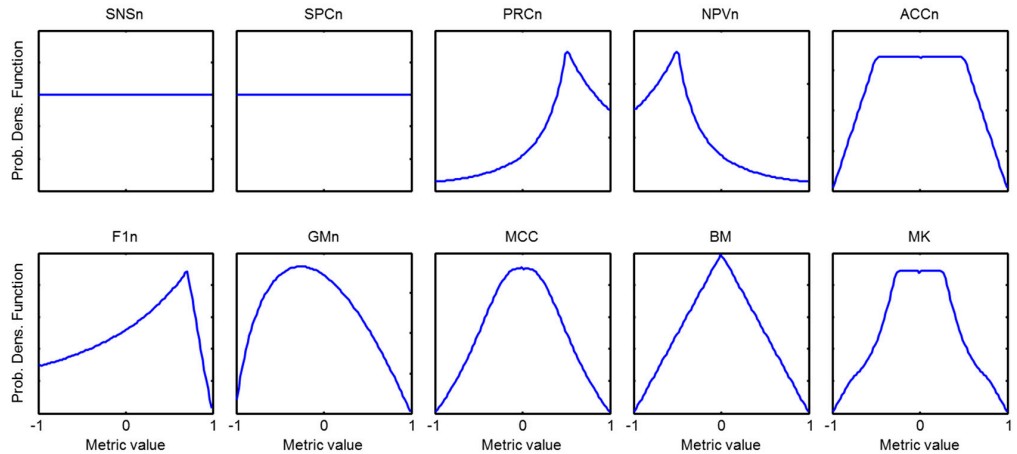

**Figure 32.** Local probability density function of every metric and $\delta = 0$.

This result can be generalized for various values of the imbalance coefficient $\delta^B$ by obtaining the $pdf(\mu_j^k, \delta^B)$ depicted in Figure 33 as a set of heatmap plots. In every plot, the horizontal axis represents the imbalance coefficient while the value of the metric is drawn in the vertical axis. The value of the $pdf$ is colour-coded.

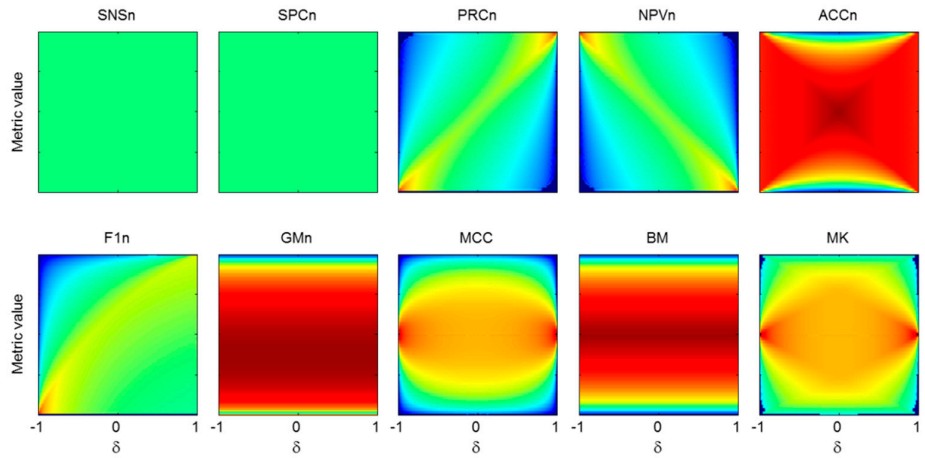

**Figure 33.** Local probability density function of every metric as a function of $\delta$. The value of pdf is colour coded.

In Figures 32 and 33, the symmetry of the statistical descriptions of the metrics can easily be observed. However, in order to achieve a more precise insight, the local skewness $\xi_j^B$ of every $pdf$ is obtained in accordance with Equation (23) and its value $\xi_j^B\left(\delta^B\right)$ is shown in Figure 34 for every metric. It can be observed that 6 metrics ($SNSn$, $SPCn$, $ACCn$, $MCC$, $BM$ *and* $MK$) have a symmetric $pdf$; one metric ($GMn$) has a $pdf$ slightly asymmetric but its asymmetry does not depend on $\delta^B$; 2 metrics ($PRCn$ and $NPVn$) have a clearly asymmetric $pdf$ but their skewness is symmetric with respect to the origin; and finally, the $F_1n$ metric has a $pdf$ and a skewness that are both asymmetric.

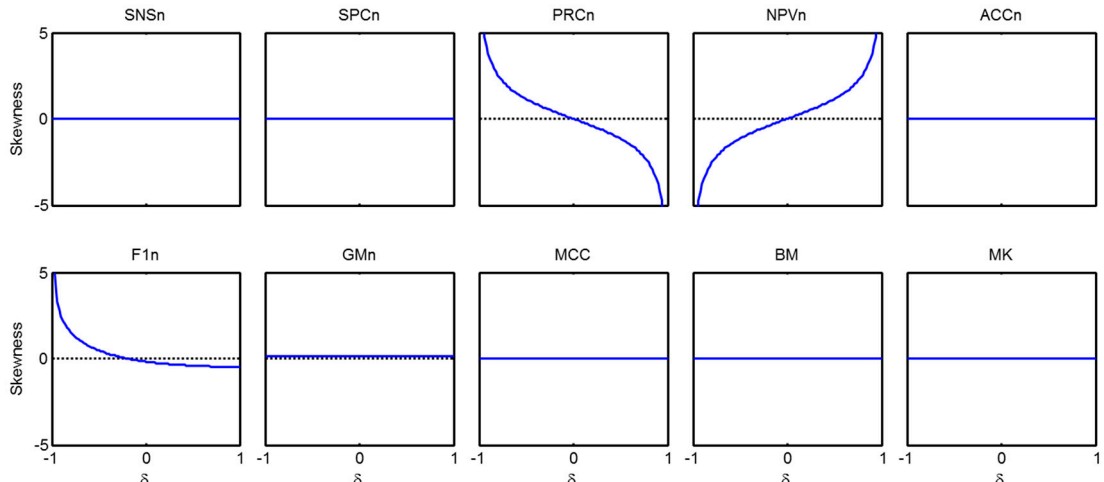

**Figure 34.** Skewness of the statistical description for every metric as a function of $\delta$.

Let us now examine the $\mu_j^B$ for all the values of the imbalance coefficient $\delta^B$, that is, $\mu_j^B\left(\alpha^B, \beta^B, \delta^B\right)$ and obtain its probability density function (pdf) which will be called global pdf (as it is obtained for every $\delta^B$). The resulting $pdf(\mu_j^k)$ is shown in Figure 35 for every metric.

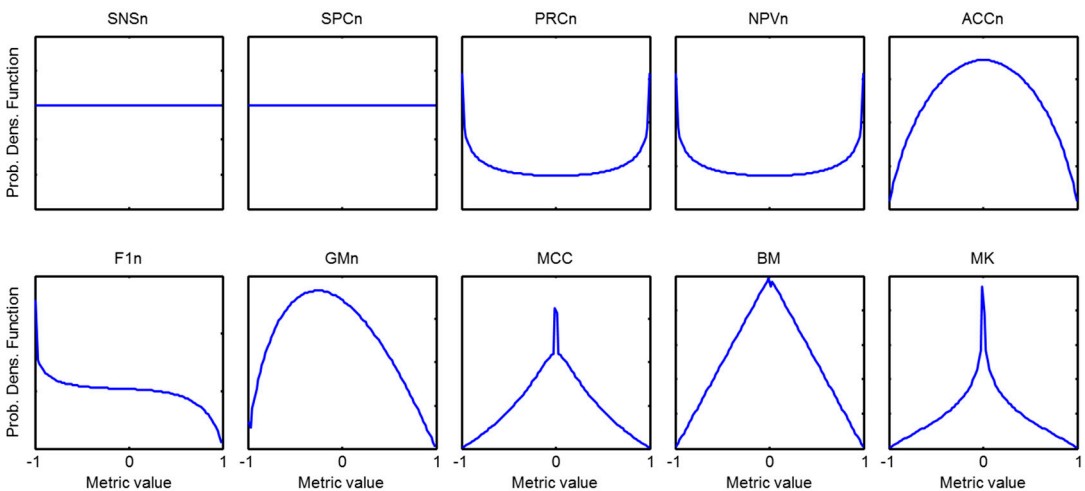

**Figure 35.** Global probability density function of every metric and $\delta = 0$.

It can be observed that all the metrics show a symmetric $pdf$ except for $GMn$ and $F_1n$. The global pdf for $GMn$ maintains the slight asymmetry of local pdf (global skewness of 0.18) since $GMn$ does not depend on $\delta$. In the cases of $PRCn$ and $NPVn$, the symmetry of the local skewness compensates for their values and hence they show a symmetric global pdf. Finally, the positive values of $F_1n$ local skewness partially compensate for its negative values (see Figure 34), which results in an almost uniform global pdf except for their extreme values (global skewness of 0.14). These results are summarized in Table 14.

**Table 14.** Summary of statistical symmetry.

| Metric | Statistical Symmetry | |
| | Local | Global (Skewness) |
|---|---|---|
| $SNSn$ | ✓ | ✓ |
| $SPCn$ | ✓ | ✓ |
| $PRCn$ | | ✓ |
| $NPVn$ | | ✓ |
| $ACCn$ | ✓ | ✓ |
| $F_1n$ | | (0.14) |
| $GMn$ | | (0.18) |
| $MCC$ | ✓ | ✓ |
| $BM$ | ✓ | ✓ |
| $MK$ | ✓ | ✓ |

## 4. Discussion

From the previous results, summarized in Tables 10, 13 and 14, it can be seen that although several thousands of combined transformations have been tested, the performance metrics only present three types of symmetries: under labelling inversion; under scoring inversion; and under full inversion (the sequence of labelling and scoring inversion).

For a certain performance metric to be symmetric under labelling inversion means that it pays attention to or focuses on, positive and negative classes with the same intensity and therefore classes can be exchanged without affecting the value of the metric. These metrics should be used in applications where the cost of misclassification is the same for each class. This is the case for 5 out of the 10 metrics tested: $ACCn$, $MCC$, $BM$, $MK$ and $GMn$.

Other metrics, however, are more focused on the classification results obtained for the positive class. This is the case of 3 metrics: $SNSn$, which only depends on $\alpha$; $PRCn$, which measures the ratio of success on the elements classified as positive; and the $F_1$ score, which is a combination of $SNSn$ and $PRCn$. These metrics found their main applications when the cost of misclassifying the positive class is higher than the cost of misclassifying the negative class, for instance, in the case of disease detection in medical diagnostics. Finally, other metrics are more focused on the classification results obtained for the negative class. This is the case of 2 metrics: $SPCn$, which only depends on $\beta$; and $NPVn$, which measures the ratio of success on the elements classified as negative. These 2 metrics are mainly applied if the most important issue is the misclassification of negative classes, for instance, in the case of identification of non-reliable clients in granting loans.

On the other hand, if a metric shows symmetric behaviour under scoring inversion it means that the good classifiers are positively scored to the same extent as bad classifiers are negatively scored. For instance, let us consider a first classifier which correctly classifies 80% of positive elements and also 70% of negative elements. Additionally, a second classifier obtains a ratio of 20% for positive and 30% for negative elements. A scoring-inversion symmetric-performance metric would have a value of, for example, +0.5 for the first classifier and a value of −0.5 for the second classifier. Therefore, the scoring symmetry indicates the relative importance assigned by the metric to the good and bad classifiers. This is the case for 6 out of the 10 metrics tested: $ACCn$, $MCC$, $BM$, $MK$, $GMn$, $SNSn$ and $SPCn$. Conversely, $GMn$ is more demanding as regards scoring good results than scoring bad results. This feature can be useful if the objective of the classification is focused on obtaining excellent results (and not just good results). Finally, on 3 of the metrics tested ($PRCn$, $NPVn$ and $F_1n$), awarding good results differs from scoring bad results in that it depends on the relative values of the parameters ($\alpha$, $\beta$ and $\delta$).

Additionally, it can be seen that metrics showing both labelling and scoring symmetries also show symmetry for the full inversion (concatenation of the two symmetries). This is the case for 4 out of the 10 metrics tested: $ACCn$, $MCC$, $BM$ and $MK$. An interesting result is that for $PRCn$ and

*NPVn*, although they have no labelling nor scoring symmetry, they do have full inversion symmetry. This fact means that swapping the positive and negative class labels also inverts how the good and bad classifiers are scored. An example of all these symmetries can be found in Table 15.

**Table 15.** Examples of symmetric behaviour of metrics under several transformations (for balanced classes). Numbers in bold represent cases of asymmetric behaviour.

| Metric | Baseline $\alpha$:0.8 ; $\beta$:0.7 | Labelling Inversion $\alpha$:0.7; $\beta$:0.8 | Scoring Inversion $\alpha$:0.2; $\beta$:0.3 | Full Inversion $\alpha$:0.3 ; $\beta$:0.2 |
|---|---|---|---|---|
| *ACCn* | 0.500 | 0.500 | −0.500 | −0.500 |
| *MCC* | 0.503 | 0.503 | −0.503 | −0.503 |
| *BM* | 0.500 | 0.500 | −0.500 | −0.500 |
| *MK* | 0.505 | 0.505 | −0.505 | −0.505 |
| *GMn* | 0.497 | 0.497 | **−0.510** | **−0.510** |
| *SNSn* | 0.600 | **0.400** | −0.600 | **−0.400** |
| *SPCn* | 0.400 | **0.600** | −0.400 | **−0.600** |
| *PRCn* | 0.455 | **0.556** | **−0.556** | −0.455 |
| *NPVn* | 0.556 | **0.455** | **−0.455** | −0.566 |
| $F_1 n$ | 0.524 | **0.474** | **−0.579** | **−0.429** |

A particular degenerate case of symmetry arises when a metric depends on none of the variables. For example, from the results obtained in this research, several metrics have shown themselves to be independent of the imbalance coefficient $\delta$. This is the case for 4 out of the 10 metrics tested: *SNSn*, *SPCn*, *GMn* and *BM*. This is a particularly interesting result, since these metrics have no kind of bias if the classes are imbalanced. Conversely, the interpretation of classification metrics which do depend on $\delta$ should be carefully considered since they can be misleading as to what a good classifier is.

Additionally, some other metrics appear to be independent from the classification success ratios: *SNSn*, which only depends on $\alpha$; and *SPCn*, which only depends on $\beta$. This can be interpreted as a sort of one-dimensionality of these metrics, that is, *SNSn* is only focused on the positive class, while *SPCn* is only concerned about the negative class.

On the other hand, the two pairs of cross-symmetries found can be straightforwardly interpreted: when the labelling of classes are inverted, *SNSn* becomes *SPCn* and *PRCn* becomes *NPVn*. Moreover, by exchanging the scoring procedure of good and bad classifiers, *PRCn* becomes *NPVn*.

Let us now focus on the interpretation of the results of statistical symmetries. Statistical local symmetry means that, for a certain dataset, that is, for a certain value of the imbalance coefficient, the probability that a random classifier obtains a good score is the same as the probability that it obtains a bad score. This is the case for 6 out of the 10 metrics tested: *ACCn, MCC, BM, MK, GMn, SNSn* and *SPCn*. They coincide with the metrics in that they have scoring symmetry, which shows that both concepts are closely related. Conversely, *GMn* has a greater probability of having a bad result than a good result, which is consistent with the fact that it is more demanding on obtaining excellent results (and not just good results). Additionally, *PRCn* obtains good results with a higher probability (lower probability in the case of *NPVn*) if the positive class is the majority class and vice versa if it is the minority class. Awarding good results differs from scoring bad ones in a way that depends on the relative values of the parameters ($\alpha$, $\beta$ and $\delta$). Finally, in the case of balanced classes, the probability of obtaining good $F_1 n$ scores is greater than obtaining bad scores for, which shows some sort of indulgent judgment. However, the detailed behaviour of $F_1 n$ scores for different values of $\delta$ is more complex.

On the other hand, statistical global symmetry means that the probability that a random classifier operating on a random dataset obtains a good score is the same as obtaining a bad score. This is the case for 8 out of the 10 metrics tested: *ACCn, MCC, BM, MK, GMn, SNSn, SPCn, PRCn* and *NPVn*. Conversely, *GMn* and $F_1 n$ are more likely to have a bad result than a good result, which can be interpreted as meaning that they are slightly tough judges.

On considering all these results and their meanings, the ten metrics can be organized into 5 clusters that show the features described in Table 16.

**Table 16.** Summary of symmetric behaviour.

| Cluster | | Metric | Independent of | | | Symmetry (under Inversion of) | | | Statistical Symmetry | |
|---|---|---|---|---|---|---|---|---|---|---|
| | | | $\alpha$ | $\beta$ | $\delta$ | Labelling | Scoring | Full | Local | Global (Skewness) |
| I | a | ACCn | | | | ✓ | ✓ | ✓ | ✓ | ✓ |
| | | MCC | | | | ✓ | ✓ | ✓ | ✓ | ✓ |
| | | MK | | | | ✓ | ✓ | ✓ | ✓ | ✓ |
| | b | BM | | | ✓ | ✓ | ✓ | ✓ | ✓ | ✓ |
| II | | SNSn | | ✓ | ✓ | SPCn | ✓ | SPCn | ✓ | ✓ |
| | | SPCn | ✓ | | ✓ | SNSn | ✓ | SNSn | ✓ | ✓ |
| III | | PRCn | | | | NPVn | NPVn | ✓ | | ✓ |
| | | NPVn | | | | PRCn | PRCn | ✓ | | ✓ |
| IV | | GMn | | | ✓ | ✓ | | | | (0.18) |
| V | | $F_1n$ | | | | | | | | (0.14) |

In Table 16, the identification of clusters has been carried out by means of informal reasoning. To formalize these analyses, every metric has been described with a set of features corresponding to the columns in Table 16. Most of the columns are binary valued (yes or no), while others admit several values. For instance, labelling symmetry value can be yes, no, $SNSn - SPCn$ cross-symmetry or $PRCn - NPVn$ cross-symmetry. In these cases, a one-hot coding mechanism (also called 1-of-K scheme) is employed [39]. The result is that each metric is defined using a set of 14 features. Although regular or advanced clustering techniques can be used [40–43], the reduced number of elements in the dataset (10 performance metrics) invites to address the problem using more intuitive methods. Using Principal Component Analysis (PCA) [44], the problem can be reduced to a bi-dimensional plane and its result is depicted in Figure 36. The 5 clusters mentioned in this section clearly appear therein.

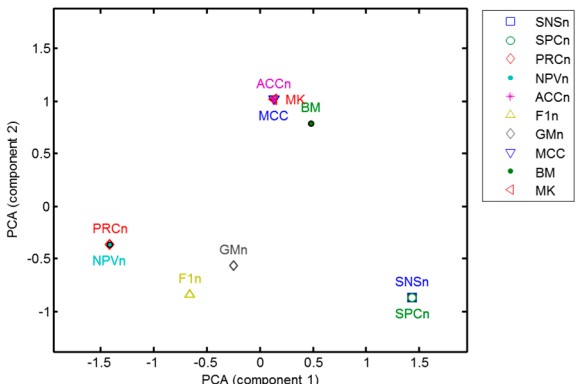

**Figure 36.** Bi-dimensional representation of performance metrics according to their symmetries.

Another way to represent how performance metrics are grouped according to their symmetries is by drawing a dendrogram [45]. To this end, the 14 features are employed to characterize each performance metric. The distances between the metrics are then computed in the space of the $\mathbb{R}^{14}$ features. These distances are employed to gauge how much the metrics are separated, as shown in Figure 37. Once again, this result is consistent with the 5 previously identified clusters.

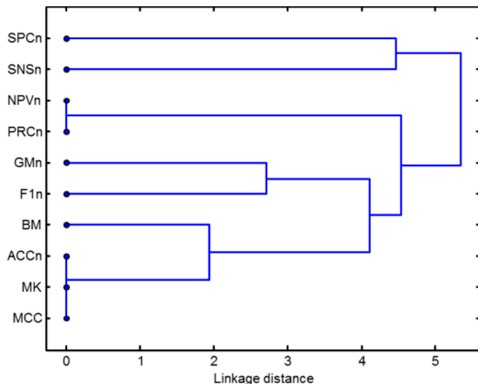

**Figure 37.** Dendrogram of performance metrics according to their symmetries.

## 5. Conclusions

Based on the results obtained in our analysis, it can be stated that the majority of the most commonly used classification performance metrics present some type of symmetry. We have identified 3 and only 3 types of symmetric behaviour: labelling inversion, scoring inversion and the combination of the two inversions. Additionally, several metrics have been revealed as being robust under imbalanced datasets, while others do not show this important feature. Finally two metrics has been identified as one-dimensional, in that they focus exclusively on the positive (sensitivity) or on the negative class (specificity). The metrics have been grouped into 5 clusters according to their symmetries.

Selecting one performance metric or another is mainly a matter of its application, depending on issues such as whether the dataset is balanced, misclassification has the same cost in either class and whether good scores should only be reserved for very good classification ratios. None of the studied metrics can be universally applied. However, according to their symmetries, two of these metrics appear especially worthy in general-purpose applications: the Bookmaker Informedness (*BM*) and the Geometric Mean (*GM*). Both of these metrics are robust under imbalanced datasets and treat both classes in the same way (labelling symmetry). The former metric (*BM*) also has scoring symmetry while the latter (*GM*) is slightly more demanding in terms of scoring good results over bad results.

In future research, the methodology for the analysis of symmetry developed in this paper can be extended to other classification performance metrics, such as those derived from multiclass confusion matrix or some ranking metrics (i.e. Receiver Operating Characteristic curve).

**Author Contributions:** A.L. conceived and designed the experiments; A.L., A.C., A.M. and J.R.L. performed the experiments, analysed the data and wrote the paper.

**Funding:** This research was funded by the Telefónica Chair "Intelligence in Networks" of the University of Seville.

**Conflicts of Interest:** The authors declare there to be no conflict of interest. The founding sponsors played no role: in the design of the study; in the collection, analyses or interpretation of data; in the writing of the manuscript; and in the decision to publish the results.

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
