# Peer review of "Exploring Symmetry of Binary Classification Performance Metrics"

_symmetry, doi:10.3390/sym11010047_

Round 1
Reviewer 1 Report
The paper studies the metric learning for binary classification. The authors systematically investigate the symmetric properties regarding the metrics, such as labelling inversion, scoring inversion and combination of the two above. They also study the impact of the imbalance property, which will largely decide the data set selection.
The paper is clearly written and easy to follow.
1. One major concern is that the authors need to discuss the latest related work over both metric learning, cross-metrics, classification even binary code learning. To name a few below:
* Unsupervised metric fusion over multiview data by graph random walk-based cross-view diffusion. IEEE Transactions on Neural Networks and Learning Systems, 28 (1), 57-70.
* Robust subspace clustering for multi-view data by exploiting correlation consensus. IEEE Transactions on Image Processing, 24(11):3939-3949, 2015.
* Cycle-Consistent Deep Generative Hashing for Cross-Modal Retrieval. IEEE Transactions on Image Processing, 2018.
2. Besides, the authors need to make up the introduction regarding Table 4, for different codes, what are their specific explanations. Please help address in the revised version.
3. The experimental analysis are encouraged to be extensive a little bit for convincing.
Author Response
The authors would like to thank the reviewer for his/her comments.
A revision of the paper has been produced in accordance with all the comments given by the reviewers.
Please find attached a point-by-point response to every comment.

Reviewer 2 Report
The paper explores role of symmetries in the context of performance metrics in binary classification. The paper is well written. The following points should be considered:
1. The authors should include references of recent literature in the field. Most of the references are prior to 2015. For example:
k-Means clustering with a new divergence-based distance metric: Convergence and performance analysis
2. The proposed method must be compared with recent work in the literature.
Author Response

(The authors gave the same response as above.)

Reviewer 3 Report
Please remove selfictations to Symmetry.
You can cite Symmetry in other journals from JCR. Thomson Reuters sees Selfcitations as cheating.
other comments:
1) Figures should have better quality.
2) Please add SI units (if any)
3) Why the topic is essential. Please add image of application.
4) The authors should cite new references (2016-2019 Web of Science).
At least new 25 referneces.
Please show that you have new knowledge.
for example
about metrics/classification methods/Euclidean distance/Manhattan distance
Fault diagnosis of single-phase induction motor based on acoustic signals
By:Glowacz, A (Glowacz, Adam)[ 1 ]
MECHANICAL SYSTEMS AND SIGNAL PROCESSING
Volume: 117 Pages: 65-80
DOI: 10.1016/j.ymssp.2018.07.044
Published:FEB 15 2019
Glowacz A.: Acoustic-Based Fault Diagnosis of Commutator Motor, ELECTRONICS, 7 (11), 299, 2018.
https://doi.org/10.3390/electronics7110299
perhaps:
k-Means clustering with a new divergence-based distance metric: Convergence and performance analysis
By:Chakraborty, S (Chakraborty, Saptarshi)[ 1 ] ; Das, S (Das, Swagatam)[ 2 ]
PATTERN RECOGNITION LETTERS
Volume: 100 Pages: 67-73
DOI: 10.1016/j.patrec.2017.09.025
Published:DEC 1 2017
Distance Metric Based Oversampling Method for Bioinformatics and Performance Evaluation
By:Tsai, MF (Tsai, Meng-Fong)[ 1 ] ; Yu, SS (Yu, Shyr-Shen)[ 1 ]
JOURNAL OF MEDICAL SYSTEMS
Volume: 40 Issue: 7
Article Number: 159
DOI: 10.1007/s10916-016-0516-3
Published:JUL 2016
Author Response

(The authors gave the same response as above.)

Round 2
Reviewer 1 Report
The authors have addressed my comments, I am satisfied with the response, and recommend its acceptance.
Reviewer 2 Report
The authors have incorporated the required changes. The paper can not be published.
Reviewer 3 Report
Thank you for responses.
Section results. It is good idea to add block diagram of performed results.
Fonts of images can be bigger.